# Training Neural Networks is NP-Hard
# in Fixed Dimension

**Vincent Froese**
Algorithmics and Computational Complexity
Faculty IV, TU Berlin
Berlin, Germany
`vincent.froese@tu-berlin.de`

**Christoph Hertrich**[*]
Department of Mathematics
London School of Economics and Political Science
London, UK
`c.hertrich@lse.ac.uk`

## Abstract

We study the parameterized complexity of training two-layer neural networks with respect to the dimension of the input data and the number of hidden neurons, considering ReLU and linear threshold activation functions. Albeit the computational complexity of these problems has been studied numerous times in recent years, several questions are still open. We answer questions by Arora et al. [2018, ICLR] and Khalife and Basu [2022, IPCO] showing that both problems are NP-hard for two dimensions, which excludes any polynomial-time algorithm for constant dimension. We also answer a question by Froese et al. [2022, JAIR] proving W[1]-hardness for four ReLUs (or two linear threshold neurons) with zero training error. Finally, in the ReLU case, we show fixed-parameter tractability for the combined parameter number of dimensions and number of ReLUs if the network is assumed to compute a convex map. Our results settle the complexity status regarding these parameters almost completely.

## 1 Introduction

Neural networks with rectified linear unit (ReLU) activations are arguably one of the most fundamental models in modern machine learning [Arora et al., 2018, Glorot et al., 2011, LeCun et al., 2015]. To use them as predictors on unseen data, one usually first fixes an architecture (the graph of the neural network) and then optimizes the weights and biases such that the network performs well on some known training data, with the hope that it will then also generalize well to unseen test data. While the ultimate goal in applications is generalization, *empirical risk minimization* (that is, optimizing the training error) is an important step in this pipeline and understanding its computational complexity is crucial to advance the theoretical foundations of deep learning.

In this paper, we aim to understand how the choice of different meta-parameters, like the input dimension and the width of the neural network, influences the computational complexity of the training problem. To this end, we focus on two-layer neural networks, which can be seen as the standard building block also for deeper architectures.

---

[*]Moved to Université Libre de Bruxelles, Belgium, and Goethe-Universität Frankfurt, Germany, after submission of this article.

37th Conference on Neural Information Processing Systems (NeurIPS 2023).

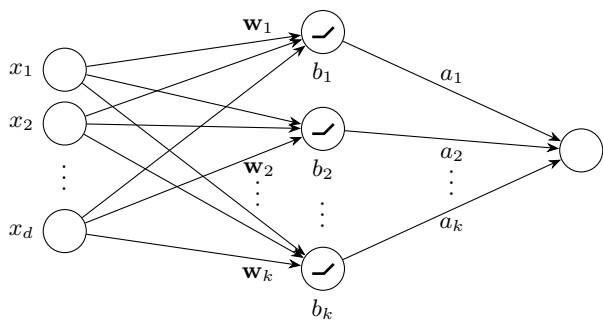

Figure 1: Neural network architecture we study in this paper: After the input layer (left) with $d$ input neurons, we have one hidden layer with $k$ ReLU neurons and a single output neuron without additional activation function.

Formally, a two-layer neural network (see Figure 1) with $d$ input neurons, $k$ hidden ReLU neurons, and a single output neuron computes a map

$$\phi\colon \mathbb{R}^d \to \mathbb{R}, \quad \phi(\mathbf{x}) = \sum_{j=1}^{k} a_j [\mathbf{w}_j \cdot \mathbf{x} + b_j]_+,$$

where $\mathbf{w}_j \in \mathbb{R}^d$ and $a_j \in \{-1, 1\}$ are the weights between the layers, $b_j$ are the biases at the hidden neurons, and $[x]_+ := \max(0, x)$ is the *rectifier* function. Notice that restricting $a_j$ to $\{-1, 1\}$ is without loss of generality because we can normalize by pulling any nonnegative factor into $\mathbf{w}_j$ and $b_j$. We also study neural networks with *linear threshold activation* in Section 5.

Given training data $\mathbf{x}_1, \ldots, \mathbf{x}_n \in \mathbb{R}^d$ with labels $y_1, \ldots, y_n \in \mathbb{R}$, the task of training such a network is to find $\mathbf{w}_j, b_j$, and $a_j$ for each $j \in [k]$ such that the training error $\sum_{i=1}^{n} \mathcal{L}(\phi(\mathbf{x}_i), y_i)$ for a given loss function $\mathcal{L}\colon \mathbb{R} \times \mathbb{R} \to \mathbb{R}_{\geq 0}$ is minimized. Throughout this work, we assume $\mathcal{L}\colon \mathbb{R} \times \mathbb{R} \to \mathbb{R}_{\geq 0}$ to be any loss function with $\mathcal{L}(x, y) = 0 \iff x = y$. Formally, the decision version of two-layer ReLU neural network training is defined as follows:

2L-RELU-NN-TRAIN($\mathcal{L}$)

**Input:** Data points $(\mathbf{x}_1, y_1), \ldots, (\mathbf{x}_n, y_n) \in \mathbb{R}^d \times \mathbb{R}$, a number $k \in \mathbb{N}$ of ReLUs, and a target error $\gamma \in \mathbb{R}_{\geq 0}$.

**Question:** Are there weights $\mathbf{w}_1, \ldots, \mathbf{w}_k \in \mathbb{R}^d$, biases $b_1, \ldots, b_k \in \mathbb{R}$, and coefficients $a_1, \ldots, a_k \in \{-1, 1\}$ such that

$$\sum_{i=1}^{n} \mathcal{L}\left( \sum_{j=1}^{k} a_j [\mathbf{w}_j \cdot \mathbf{x}_i + b_j]_+, \; y_i \right) \leq \gamma?$$

Note that in the over-parameterized case where $k \geq n$, the network can exactly fit any $n$ input points[2] (achieving training error $\gamma = 0$) [Zhang et al., 2021, Theorem 1]. Thus, we henceforth assume that $k < n$.

2L-RELU-NN-TRAIN($\mathcal{L}$) is known to be NP-hard [Dey et al., 2020, Goel et al., 2021], but all known reductions require the input dimension to be part of the input. The current state-of-the-art exact algorithm for convex loss $\mathcal{L}$ is by Arora et al. [2018] and runs in $O(2^k n^{dk} \operatorname{poly}(L))$ time, where $L$ is the input bit-length.

As regards the computational complexity of 2L-RELU-NN-TRAIN($\mathcal{L}$), Arora et al. [2018] posed the question(s) whether a running time

> *"that is polynomial in the data size and/or the number of hidden nodes, assuming that the input dimension is a fixed constant"*

is possible. That is, they asked two questions. The first corresponds to the "and" statement, which can be phrased as follows:

---

[2]Assuming that $y_i = y_j$ whenever $\mathbf{x}_i = \mathbf{x}_j$.

**Question 1**: Is there an algorithm running in $(nk)^{f(d)} \operatorname{poly}(L)$ time for some function $f$?

In other words, the question is whether 2L-ReLU-NN-Train($\mathcal{L}$) is in the complexity class XP when parameterized by $d$. The second question corresponding to the "or" statement can then be interpreted as

**Question 2**: Is there an algorithm running in $n^{f(d)}g(k,d) \operatorname{poly}(L)$ or $k^{f(d)}g(n,d) \operatorname{poly}(L)$ time for some functions $f$ and $g$?

We observe that the second running time is clearly possible since $k < n$ holds by assumption, and hence the algorithm by Arora et al. [2018] runs in $g(n,d) \operatorname{poly}(L)$ time. Hence, it remains open whether $n^{f(d)}g(k,d) \operatorname{poly}(L)$ time is possible, which is equivalent to (uniform) fixed-parameter tractability with respect to $k$ for every constant $d$.

Clearly, Question 1 is the stronger statement, that is, a positive answer implies a positive answer to Question 2. Arora et al. [2018] conclude with

> *"Resolving this dependence on network size would be another step towards clarifying the theoretical complexity of training ReLU DNNs and is a good open question for future research, in our opinion."*

Note that Froese et al. [2022] proved that, for $k = 1$, there is no algorithm running in $g(d)n^{o(d)}$ time unless the Exponential Time Hypothesis fails. Hence, this result already partially answered the two questions above by excluding any algorithm running in $n^{o(d)}g(d,k) \operatorname{poly}(L)$ time.

**Our Contribution.** In this paper, we answer Question 1 negatively by showing NP-hardness for $d = 2$ in Theorem 1, indicating that we cannot get rid of the exponential dependence on the network size in the algorithm by Arora et al. [2018] even if the dimension is fixed. As regards Question 2, we further exclude (assuming the Exponential Time Hypothesis) any algorithm running in time $n^{o(d)}g(d,k) \operatorname{poly}(L)$ even for the case $\gamma = 0$ and prove W[1]-hardness with respect to $d$ for $k = 4$ (Theorem 6), which answers an open question by Froese et al. [2022].

We also obtain analogous hardness results if linear threshold activation functions are used instead of ReLUs. As in the ReLU case, it is well-known that training linear threshold networks is NP-hard [Blum and Rivest, 1992, Khalife and Basu, 2022]. The running time of the state-of-the-art algorithm due to Khalife and Basu [2022] is polynomial in $n$ for fixed $d$ and $k$, but exponential in the latter two parameters. Khalife and Basu [2022] posed an analogous question to Question 1 for linear threshold networks, which we answer negatively in Corollary 7, excluding a polynomial running time even for fixed dimension. We also show that we cannot expect fixed-parameter tractability with respect to $d$ even for $k = 1$ (Corollary 8) and also not for $k = 2$ and $\gamma = 0$ (Corollary 9).

On the positive side, we give an algorithm running in $2^{O(k^2 d)} \operatorname{poly}(k, L)$ time for ReLU neural networks if $\gamma = 0$ and the function computed by the network is assumed to be convex (Theorem 10). Note that this running time yields fixed-parameter tractability with respect to $k$ for every constant $d$, and thus answers Question 2 positively for this restricted special case. We would like to emphasize that we do not expect this algorithm to have any practical relevance. We rather see its value in being a puzzle piece for understanding the computational complexity of the training problem.

**Implications and Limitations.** In the following we provide a brief discussion of the implications and limitations of our results from various perspectives.

*Input Dimension.* Theorem 1 implies that 2L-ReLU-NN-Train($\mathcal{L}$) is in fact NP-hard for every fixed $d \geq 2$. The straight-forward reduction simply pads all the input vectors with $d - 2$ zeros. Similarly, Corollary 7 holds for every fixed $d \geq 2$.

*Target Error.* The hardness results Theorems 1 and 6 and Corollaries 7 and 9 also hold for every fixed $\gamma \geq 0$. The reduction is straight-forward by introducing a set of incompatible data points which force the network to incur an additional error of $\gamma$. For our positive result Theorem 10, however, there is indeed a difference in the complexity between the two cases $\gamma = 0$ and $\gamma > 0$. While we show fixed-parameter tractability for $\gamma = 0$, the same problem is W[1]-hard for $\gamma > 0$, already in the case $k = 1$ [Froese et al., 2022].

*Number of ReLUs.* It is not too difficult to see (although it requires some work) that our particular reduction in Theorem 6 can be extended to any $k \geq 4$ by introducing more data points far away from the existing data points which enforce the usage of additional ReLUs which then cannot be used to fit the data points of the actual reduction. Therefore, Theorem 6 holds for every fixed $k \geq 4$. Similarly, Corollary 8 holds for every fixed $k \geq 1$ and Corollary 9 holds for every fixed $k \geq 2$.

*Other Activation Functions.* Our hardness results hold for the piecewise linear ReLU activation function and the piecewise constant linear threshold activation function. Extending them to other piecewise linear or constant activation functions like leaky ReLU or maxout should be straight-forward. However, achieving analogous results for smooth activation functions like sigmoids probably requires fundamentally different techniques and is beyond the scope of this paper.

*Training vs. Learning.* Our results are concerned with the problem of minimizing the training error. While this is inherently different from minimizing the generalization error, there are indeed deep connections between these two problems [Shalev-Shwartz and Ben-David, 2014]. In particular, as pointed out by Goel et al. [2021], hardness of training implies hardness of proper learning if one permits arbitrary data distributions. However, such hardness results can often be overcome by either posing additional assumptions on the data distributions or switching to more general learning paradigms like improper learning [Goel et al., 2017].

*Exact vs. Approximate Training.* In practice, it arguably often suffices to train a neural network to approximate instead of exact optimality. The results in this paper are concerned with solving the training problem to exact global optimality. However, since Theorems 1 and 6 and Corollaries 7 and 9 already hold for training error $\gamma = 0$, they even rule out the existence of approximation algorithms with any multiplicative factor. We conceive that for appropriate notions of *additive* approximation (see, e.g., Goel et al. [2021]), our reductions can also be used to show hardness of additive approximation. However, this would significantly increase the technical complexity of the analysis and is therefore beyond the scope of this paper. We leave it as an open research question to analyze the influence of meta-parameters like input dimension and number of hidden neurons on additive approximation of the training problem.

**Related Work.** Dey et al. [2020] and Goel et al. [2021] showed NP-hardness of 2L-RELU-NN-TRAIN($\mathcal{L}$) for $k = 1$, but require non-constant dimension. For target error $\gamma = 0$, the problem is NP-hard for every constant $k \geq 2$ and polynomial-time solvable for $k = 1$ [Goel et al., 2021]. Goel et al. [2021] provide further conditional running time lower bounds and (additive) approximation hardness results. Froese et al. [2022] considered the parameterized complexity regarding the input dimension $d$ and proved W[1]-hardness and an ETH-based running time lower bound of $n^{\Omega(d)}$ for $k = 1$.

Boob et al. [2022] studied networks where the output neuron also is a ReLU and proved NP-hardness (and implicitly W[1]-hardness with respect to $d$) for $k = 2$ and $\gamma = 0$. Bertschinger et al. [2022] showed that training 2-layer ReLU networks with two output and two input neurons ($\mathbb{R}^2 \to \mathbb{R}^2$) is complete for the class $\exists \mathbb{R}$ (existential theory of the reals) and thus likely not contained in NP. This also implies NP-hardness, but note that in contrast to our results, their reduction does not work for one-dimensional outputs. Their result strengthens a previous result by Abrahamsen et al. [2021] who proved $\exists \mathbb{R}$-completeness for networks with a specific (not fully connected) architecture. Pilanci and Ergen [2020] showed that training 2-layer neural networks can be formulated as a convex program which yields a polynomial-time algorithm for constant dimension $d$. However, they considered a regularized objective and their result requires the number $k$ of hidden neurons to be very large (possibly equal to the number $n$ of input points) and hence does not contradict our NP-hardness result for $d = 2$.

To study the computational complexity of training ReLU networks, a crucial ingredient is to know the set of (continuous and piecewise linear) functions precisely representable with a certain network architecture. This is well-understood for two-layer networks [Arora et al., 2018, Bertschinger et al., 2022, Dereich and Kassing, 2022], but much trickier for deeper networks [Haase et al., 2023, Hertrich and Sering, 2023, Hertrich et al., 2021]. Similar to the study of ReLU networks by Arora et al. [2018], Khalife and Basu [2022] studied the expressiveness and training complexity for linear threshold activation functions. For an extensive survey on intersections of deep learning and polyhedral theory, we refer to Huchette et al. [2023].

## 2 Preliminaries

**Notation.** For $n \in \mathbb{N}$, we define $[n] := \{1, \ldots, n\}$. For $X \subseteq \mathbb{R}^d$, we denote by $\mathrm{aff}(X)$ the affine hull of $X$ and by $\dim(X)$ the dimension of $\mathrm{aff}(X)$.

**Parameterized Complexity.** We assume basic knowledge on computational complexity theory. Parameterized complexity is a multivariate approach to analyze the computational complexity of problems [Downey and Fellows, 2013, Cygan et al., 2015].

An instance $(x, k)$ of a parameterized problem $L \subseteq \Sigma^* \times \mathbb{N}$ is a pair with $x \in \Sigma^*$ being a problem instance and $k \in \mathbb{N}$ being the value of a certain *parameter*. A parameterized problem $L$ is *fixed-parameter tractable (fpt)* (contained in the class FPT) if there exists an algorithm deciding whether $(x, k) \in L$ in $f(k) \cdot |x|^{O(1)}$ time, where $f$ is a function solely depending on $k$. Note that a parameterized problem in FPT is polynomial-time solvable for every constant parameter value where, importantly, the degree of the polynomial does not depend on the parameter value. The class XP contains all parameterized problems which can be solved in polynomial time for constant parameter values, that is, in time $f(k) \cdot |x|^{g(k)}$. It is known that FPT $\subsetneq$ XP. The class W[1] contains parameterized problems which are widely believed not to be in FPT. That is, a W[1]-hard problem (e.g. CLIQUE parameterized by the size of the sought clique) is not solvable in $f(k) \cdot |x|^{O(1)}$ time. It is known that FPT $\subseteq$ W[1] $\subseteq$ XP.

W[1]-hardness is defined via *parameterized reductions*. A parameterized reduction from $L$ to $L'$ is an algorithm mapping an instance $(x, k)$ in $f(k) \cdot |x|^{O(1)}$ time to an instance $(x', k')$ such that $k' \leq g(k)$ for some function $g$ and $(x, k) \in L$ if and only if $(x', k') \in L'$.

**Exponential Time Hypothesis.** The Exponential Time Hypothesis (ETH) [Impagliazzo and Paturi, 2001] states that 3-SAT cannot be solved in subexponential time in the number $n$ of Boolean variables in the input formula, that is, there exists a constant $c > 0$ such that 3-SAT cannot be solved in $O(2^{cn})$ time.

The ETH implies FPT $\neq$ W[1] [Cygan et al., 2015] (which implies P $\neq$ NP). In fact, ETH implies that CLIQUE cannot be solved in $\rho(k) \cdot n^{o(k)}$ time for any function $\rho$, where $k$ is the size of the sought clique and $n$ is the number of vertices in the graph [Cygan et al., 2015].

**Geometry of 2-Layer ReLU Networks.** For proving our results, it is crucial to understand the geometry of a function $\phi \colon \mathbb{R}^d \to \mathbb{R}$ represented by a two-layer ReLU network. Here, we only discuss properties required to understand our results and refer to Arora et al. [2018], Bertschinger et al. [2022], Dereich and Kassing [2022] for additional discussions in this context. Such a function $\phi$ is a continuous and piecewise linear function. Each hidden neuron with index $j \in [k]$ defines a hyperplane $\mathbf{w}_j \cdot \mathbf{x} + b_j = 0$ in $\mathbb{R}^d$. These $k$ hyperplanes form a hyperplane arrangement. Inside each cell of this hyperplane arrangement, the function $\phi$ is affine. The graph of $\phi$ restricted to such a cell is called a *(linear) piece* of $\phi$.

Consider a hyperplane $H$ from the hyperplane arrangement. Let $\mathbf{w}$ be an orthonormal vector of $H$ and let $J \subseteq [k]$ be the non-empty subset of indices of neurons which induce precisely $H$. Note that $\mathbf{w}_j$ is a scaled version of $\mathbf{w}$ for each $j \in J$. Let $\mathbf{x} \in \mathbb{R}^d$ be a point on $H$ which does not lie on any other hyperplane in the arrangement. There are exactly two $d$-dimensional cells in the arrangement containing $\mathbf{x}$: one on each side of $H$. The difference of the directional derivatives of the corresponding two pieces of $\phi$ in the direction of $\mathbf{w}$ is exactly $\sum_{j \in J} a_j \|\mathbf{w}_j\|$. In particular, this is independent of $\mathbf{x}$ and therefore constant along $H$. If this value is positive, we say that $H$ is a *convex* hyperplane of $\phi$. If it is negative, we say that $H$ is a *concave* hyperplane of $\phi$. Note that this matches with $\phi$ being convex or concave locally around every point $\mathbf{x} \in H$ which does not belong to any other hyperplane in the arrangement. Moreover, a point $\mathbf{x} \in \mathbb{R}^d$ is called a convex (concave) *breakpoint* of $\phi$ if it lies exclusively on one convex (concave) hyperplane of $\phi$.

One important observation we will heavily use is the following: If we know that $\phi$ originates from a 2-layer neural network with $k$ hidden neurons and we know that we need indeed $k$ distinct hyperplanes to separate the pieces of $\phi$, then each hyperplane must be induced by exactly one neuron (and not several). Then the hyperplane corresponding to the $j$-th neuron is convex if and only if $a_j > 0$ and concave if and only if $a_j < 0$. For input dimension $d = 2$, each of the hyperplanes in the arrangement

is actually a line in $\mathbb{R}^2$. We call such a line a *breakline* and define *convex* and *concave* breaklines accordingly.

## 3 NP-Hardness for Two Dimensions

In this section we prove our main result that 2L-RELU-NN-TRAIN($\mathcal{L}$) is NP-hard for two dimensions, thus excluding any running time of the form $(nk)^{f(d)}$.

**Theorem 1.** 2L-RELU-NN-TRAIN($\mathcal{L}$) *is NP-hard even for $d = 2$ and $\gamma = 0$.*

We give a polynomial-time reduction from the following NP-complete problem [Garey and Johnson, 1979].

POSITIVE ONE-IN-THREE 3-SAT (POITS)
 **Input:**    A Boolean formula $F$ in conjunctive normal form with three positive literals
       per clause.
 **Question:**   Is there a truth assignment for the variables such that each clause in $F$ has
       exactly one true literal?

Our construction will be such that the function represented by the neural network is equal to zero everywhere except for a finite set of "stripes", in which the function forms a *levee* (see Definition 2), that is, when looking at a cross section, the function goes up from 0 to 1, stays constant for a while, and goes down from 1 to 0 again. See Figure 2 (right) for a top view of a levee and Figure 3 for a cross section of a levee.

**Definition 2.** *A levee with slope $s \in \mathbb{R}$ (centered at the origin) is the function $f_s \colon \mathbb{R}^2 \to \mathbb{R}$ with*

$$f_s(x_1, x_2) = \begin{cases} 0, & \text{if } |x_2 - sx_1| \geq 2, \\ 1, & \text{if } |x_2 - sx_1| \leq 1, \\ 2 + x_2 - sx_1, & \text{if } x_2 - sx_1 \in\, ]-2, -1[, \\ 2 - x_2 + sx_1, & \text{if } x_2 - sx_1 \in\, ]1, 2[. \end{cases} \tag{1}$$

**Observation 3.** *A levee $f_s$ is a continuous, piecewise linear function with two convex and two concave breaklines. It can be realized with four ReLUs as follows:*

$$f_s(x_1, x_2) = [x_2 - sx_1 + 2]_+ - [x_2 - sx_1 + 1]_+ - [x_2 - sx_1 - 1]_+ + [x_2 - sx_1 - 2]_+.$$

Similar levees have been used by Bertschinger et al. [2022] to prove $\exists\mathbb{R}$-completeness of neural network training, however, in a conceptually very different way. In their work, levees encode variable values via the *slopes of the function* on the non-constant regions of the levee. In contrast, in our reduction, we encode discrete choices via rotation of the levees, that is, via the *slopes of the breaklines in the two-dimensional input space*.

**Selection Gadget.** We describe a gadget allowing us to model a discrete choice between $\ell$ many possibilities (levees). An illustration of the selection gadget is given in Figures 2 and 3.

Each of the $\ell$ different choices corresponds to one of $\ell$ different slopes $s_1 < s_2 < \cdots < s_\ell$. The gadget consists of a total of $29 + 2\ell$ many data points with labels in $\{0, 1/3, 2/3, 1\}$. The precise position of the data points is described in Appendix A.

It is not too difficult to verify that a levee with slope $s_i$, $i \in [\ell]$, fits all data points of a selection gadget. We omit the simple but tedious calculations here. More intricately, the following lemma shows that a selection gadget indeed models a discrete choice between exactly $\ell$ possibilities. The technically involved proof of the lemma is deferred to Appendix A.

**Lemma 4.** *Let $f \colon \mathbb{R}^2 \to \mathbb{R}$ be a continuous piecewise linear function with only four breaklines that fits all the data points of the selection gadget. Then, $f = f_{s_i}$ for some $i \in [\ell]$.*

**Combining Multiple Selection Gadgets.** Having constructed and understood a single selection gadget, the next step is to use multiple of these gadgets simultaneously. To this end, we will "stack multiple selection gadgets upon each other along the $x_2$-axis". The precise construction and a formal proof of the following lemma are contained in Appendix A.

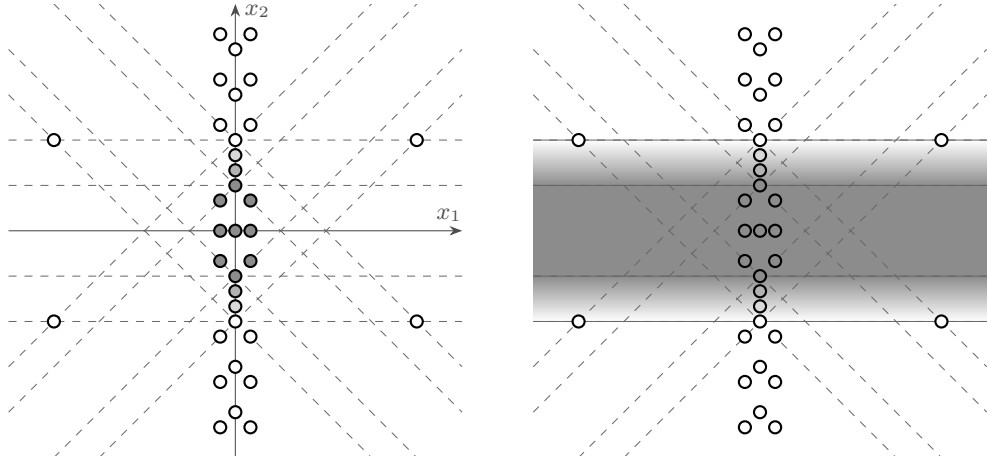

Figure 2: Illustration of the selection gadget with $\ell = 3$ and $s_1 = -1$, $s_2 = 0$, $s_3 = 1$. Both figures show the $x_1$-$x_2$-plane while the $y$-coordinate is indicated via the darkness of the gray color. The left picture shows all data points belonging to the gadget as well as the breaklines of the three possible levees fitting the data points. In addition to these features, the right picture shows a levee with slope $s_2 = 0$ as one of three possibilities to fit the data points of the gadget.

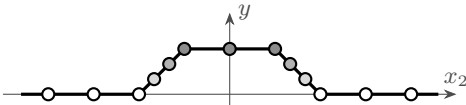

Figure 3: Cross section of the selection gadget along the $x_2$-axis. The data points of the gadget force the function $f$ to attain the shape of a levee.

**Lemma 5** (Informal version of Lemma 11). *Suppose $m$ selection gadgets are placed along the $x_2$-axis with sufficient distance to each other. Then the unique way of fitting all these gadgets with a single continuous piecewise linear function $f \colon \mathbb{R}^2 \to \mathbb{R}$ with at most $4m$ breaklines is selecting exactly one levee for each of the $m$ gadgets and summing up the corresponding functions.*

**Global Construction.** We are now ready to describe the overall layout of the reduction.

For a given formula $F = C_1 \wedge C_2 \ldots \wedge C_m$ with variables $v_1, \ldots, v_n$, we construct data points in $\mathbb{R}^2 \times \mathbb{R}$ such that they can be fitted exactly with $k = 4(m + n)$ ReLUs if and only if $F$ is a yes-instance of POITS. See Figure 4 for an illustration. Our construction consists of $m + n$ selection gadgets, namely one for each clause and one for each variable, and $3m$ further data points. Each of the $m$ selection gadgets corresponding to a clause determines which literal of this clause we choose to be true. Each of the $n$ selection gadgets corresponding to a variable determines whether this variable is true or false. The $3m$ remaining data points ensure that these choices are consistent. More precisely, if the $r$-th literal of clause $C_j$ is $v_i$, then we introduce a data point $\mathbf{p}_{j,r}$ with label $y = 1$ at the intersection of the levee corresponding to the $r$-th choice in $C_j$ and the levee corresponding to setting $v_j$ to false. This makes sure that the $r$-th literal of $C_j$ is selected if and only if $v_i$ is set to true.

*Sketch of the proof of Theorem 1.* Our detailed proof in Appendix A starts by showing that the data points $\mathbf{p}_{j,r}$ are indeed positioned such that they only interfere with the two intended levees and no other selection gadgets (Lemma 12). Then, for a given instance of POITS, we construct an instance of 2L-RELU-NN-TRAIN($\mathcal{L}$) as described above. Making use of our lemmas and Observation 3, we show that the possible solutions of POITS are in one-to-one correspondence to the possible solutions of 2L-RELU-NN-TRAIN($\mathcal{L}$), which decompose into a sum of levees. This finishes the reduction. □

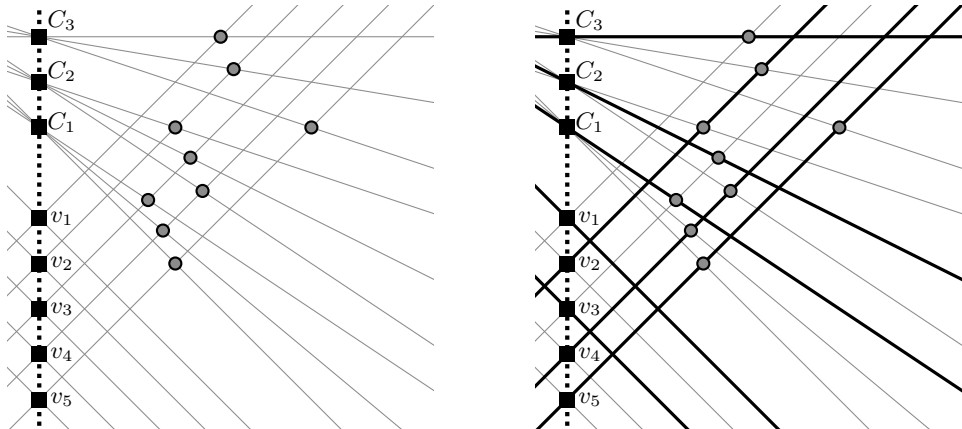

Figure 4: Global construction layout for the reduction from POITS to 2L-ReLU-NN-Train($\mathcal{L}$). The figure shows the construction for the instance $(v_5 \vee v_4 \vee v_3) \wedge (v_4 \vee v_3 \vee v_2) \wedge (v_5 \vee v_2 \vee v_1)$. The vertical dotted line is the $x_2$-axis along which we place all the selection gadgets. Each gadget is depicted with a black square. Each solid gray line depicts one possible levee. Each gray circle depicts a data point $\mathbf{p}_{j,r}$ with label one. The picture on the right additionally shows one possible solution to the given instance. Indeed, choosing levees corresponding to the solid black lines selects exactly one levee per selection gadget and exactly one levee passing through each of the nine additional data points. This corresponds to the truth assignment $v_1 = v_3 = $ true and $v_2 = v_4 = v_5 = $ false.

## 4   W[1]-Hardness for Four ReLUs

We show that fixed-parameter tractability with respect to $d$ is unlikely even for target error zero and four ReLUs. In fact, we prove a running time lower bound of $n^{\Omega(d)}$ based on the ETH.

**Theorem 6.** 2L-ReLU-NN-Train($\mathcal{L}$) *with $k = 4$ and $\gamma = 0$ is W[1]-hard with respect to $d$ and not solvable in $\rho(d)n^{o(d)} \operatorname{poly}(L)$ time (where $L$ is the input bit-length) for any function $\rho$ assuming the ETH.*

We prove Theorem 6 with a parameterized reduction from the 2-Hyperplane Separability problem (see Appendix B).

2-Hyperplane Separability
**Input:** Two point sets $Q$ and $P$ in $\mathbb{R}^d$.
**Question:** Are there two hyperplanes that strictly separate $Q$ and $P$?

Here, two hyperplanes *strictly separate* $Q$ and $P$ if, for every pair $(\mathbf{q}, \mathbf{p}) \in Q \times P$, the open line segment $\mathbf{qp}$ is intersected by at least one hyperplane and no point from $Q \cup P$ is contained in any of the two hyperplanes. Giannopoulos et al. [2009] showed that this problem is W[1]-hard with respect to $d$ and not solvable in $\rho(d)m^{o(d)} \operatorname{poly}(L)$ time assuming the ETH (where $m := |Q \cup P|$ and $L$ is the instance size). The idea of our reduction is that a separating hyperplane can be realized by two ReLUs approximating a "step function". To this end, we add some auxiliary points to $Q \cup P$ to ensure the separating behaviour of the ReLUs.

## 5   Hardness Results for Linear Threshold Activations

A nowadays less popular, but more classical activation function than ReLU is the linear threshold function $x \mapsto \mathbb{1}_{\{x > 0\}}$. We define 2L-LT-NN-Train($\mathcal{L}$) as the decision version of the training problem for linear threshold functions. It is defined exactly as 2L-ReLU-NN-Train($\mathcal{L}$) with two modifications: first, the ReLU function $[x]_+$ is replaced with $\mathbb{1}_{\{x > 0\}}$, and second, we do not assume $a_j \in \{-1, 1\}$ anymore because the normalization used in the ReLU case does not apply here. Instead we allow arbitrary $a_j \in \mathbb{R}$.

As in the ReLU case, the crucial ingredient to study the training complexity of linear threshold networks is their geometry. To this end, observe that every function represented by a 2-layer linear

threshold network is piecewise constant, where the pieces emerge from the hyperplane arrangement defined by the $k$ hyperplanes corresponding to the hidden neurons. Since our reductions for the ReLU case always use two ReLUs to approximate "step functions" from 0 to 1 and from 1 to 0, it is easy to adapt the reductions to the linear threshold case.

**Corollary 7.** 2L-LT-NN-TRAIN($\mathcal{L}$) *is NP-hard even for $d = 2$ and $\gamma = 0$.*

*Proof.* We use a reduction analogous to the one in the proof of Theorem 1. Instead of a sum of levees, we use a sum of "stripes" within which the function attains value 1. With this idea, it is straight-forward to build selection gadgets and an analogous global construction. Note that the number $k$ of required linear threshold neurons is only $k = 2(m + n)$ for a POITS instance with $m$ clauses and $n$ variables because each stripe can be realized with two linear threshold neurons instead of the four ReLUs required to build a levee. $\square$

For the sake of completeness, we note that also the W[1]-hardness result by Froese et al. [2022] extends to linear threshold functions. To this end, consider the $\ell^p$-loss $\ell^p(\hat{y}, y) = |\hat{y} - y|^p$, where $\ell^0$ simply counts the non-zero components of $\hat{y} - y$.

**Corollary 8.** *For each $p \in [0, \infty[$, 2L-LT-NN-TRAIN($\ell^p$) with $k = 1$ is NP-hard, W[1]-hard with respect to $d$ and not solvable in $\rho(d)n^{o(d)} \operatorname{poly}(L)$ time (where $L$ is the input bit-length) for any function $\rho$ assuming the ETH.*

*Proof.* Having a careful look into the reduction from MULTICOLORED CLIQUE by Froese et al. [2022], it turns out that the single ReLU neuron used in this reduction can be replaced by a linear threshold neuron without changing the logic of the reduction. $\square$

Finally, also Theorem 6 finds its analogue for the linear threshold case.

**Corollary 9.** 2L-LT-NN-TRAIN($\mathcal{L}$) *with $k = 2$ and $\gamma = 0$ is W[1]-hard with respect to $d$ and not solvable in $\rho(d)n^{o(d)} \operatorname{poly}(L)$ time (where $L$ is the input bit-length) for any function $\rho$ assuming the ETH.*

*Proof.* The proof is analogous to (even much easier than) the one of Theorem 6. Instead of two ReLUs to realize a step of height one, we can simply use one linear threshold neuron (which is why we obtain hardness already for $k = 2$ in this case). Note that we do not even need to introduce additional data points, that is, we obtain a much more direct reduction from 2-HYPERPLANE SEPARABILITY. $\square$

## 6 An Algorithm for Exact Fitting in the Convex Case

Contrasting the previous hardness results for 2L-RELU-NN-TRAIN($\mathcal{L}$), we now consider the tractable special case where all coefficients $a_j$ are 1. In this case (called 2L-RELU-NN-TRAIN($\mathcal{L}$)$^+$), the neural network realizes a convex continuous piecewise linear function $\phi$ with at most $2^k$ distinct (affine) pieces. We show fixed-parameter tractability for the parameter $d+k$ if $\gamma = 0$ (see Appendix C). Note that, for $\gamma > 0$, W[1]-hardness with respect to $d$ already holds for $k = 1$ Froese et al. [2022].

**Theorem 10.** 2L-RELU-NN-TRAIN($\mathcal{L}$)$^+$ *can be solved in $2^{O(k^2 d)} \operatorname{poly}(k, L)$ time for $\gamma = 0$, where $L$ is the input bit-length.*

The proof idea is that the convexity of $\phi$ allows for a branching algorithm assigning the input data points to the $2^k$ pieces of $\phi$. We note that Theorem 10 analogously holds for the concave case where $a_j = -1$ for all $j \in [k]$. However, if positive and negative coefficients $a_j$ are allowed, then our search tree approach does not work due to non-convexity. Indeed, Theorem 6 implies that this approach cannot work already for $k = 4$. It is unclear whether this issue can be resolved for $k = 2$ or $k = 3$.

## 7 Conclusion

We closed several gaps in the literature regarding the computational complexity of training two-layer neural networks. Our results give some insight into the geometry of functions realized by such

networks and yield a better understanding of their complexity and expressiveness. We thereby settled the border of computational tractability almost completely. The remaining open questions are the following:

- Is the problem with $d = 2$ in FPT when parameterized by $k$? This is open for both ReLUs and linear thresholds. Note that W[1]-hardness with respect to $k$ for any constant $d$ would answer Question 2 negatively.
- Is the case $\gamma = 0$ and $k \in \{2, 3\}$ in FPT with parameter $d$ for ReLUs?

In a broader context, open directions are to further study the computational complexity in appropriate approximate settings, draw further conclusions on generalization, and understand deeper network architectures.

## Acknowledgments and Disclosure of Funding

Christoph Hertrich is supported by the European Research Council (ERC) under the European Union's Horizon 2020 research and innovation programme (grant agreements ScaleOpt–757481 and ForEFront–615640).

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
