# Supplemental:
# Training Neural Networks is NP-Hard
# in Fixed Dimension

## A  Detailed Proof of NP-Hardness for Two Dimensions

In this section we provide the omitted details to prove Theorem 1. We start by describing the precise positions of the data points in the selection gadget.

First, we place 13 data points on the $x_2$-axis (with $x_1 = 0$, we call this vertical line $h_0$):

| $x_2$ | $-4$ | $-3$ | $-2$ | $-5/3$ | $-4/3$ | $-1$ | $0$ | $1$ | $4/3$ | $5/3$ | $2$ | $3$ | $4$ |
|---|---|---|---|---|---|---|---|---|---|---|---|---|---|
| $y$ | $0$ | $0$ | $0$ | $1/3$ | $2/3$ | $1$ | $1$ | $1$ | $2/3$ | $1/3$ | $0$ | $0$ | $0$ |

Next, we need a small $\epsilon > 0$ to be chosen later in a global context. The only condition we impose on $\epsilon$ to make the selection gadget work is that $\epsilon \le \min\left\{\frac{1}{3|s_1|}, \frac{1}{3|s_\ell|}\right\}$. Based on this, we place 9 data points parallel to the $x_2$-axis with $x_1 = -\epsilon$ (we call the corresponding vertical line $h_{-\epsilon}$):

| $x_2$ | $-4 - \epsilon s_\ell$ | $-3 - \epsilon s_\ell$ | $-2 - \epsilon s_\ell$ | $-1 - \epsilon s_1$ | $0$ | $1 - \epsilon s_\ell$ | $2 - \epsilon s_1$ | $3 - \epsilon s_1$ | $4 - \epsilon s_1$ |
|---|---|---|---|---|---|---|---|---|---|
| $y$ | $0$ | $0$ | $0$ | $1$ | $1$ | $1$ | $0$ | $0$ | $0$ |

Furthermore, similar to above, we place 9 data points parallel to the $x_2$-axis with $x_1 = \epsilon$ (we call the corresponding line $h_\epsilon$):

| $x_2$ | $-4 + \epsilon s_1$ | $-3 + \epsilon s_1$ | $-2 + \epsilon s_1$ | $-1 + \epsilon s_\ell$ | $0$ | $1 + \epsilon s_1$ | $2 + \epsilon s_\ell$ | $3 + \epsilon s_\ell$ | $4 + \epsilon s_\ell$ |
|---|---|---|---|---|---|---|---|---|---|
| $y$ | $0$ | $0$ | $0$ | $1$ | $1$ | $1$ | $0$ | $0$ | $0$ |

Finally, we place $2(\ell - 1)$ many data points as follows: for each $i \in [\ell - 1]$, we introduce one data point $\mathbf{q}_i^- := (-\frac{4}{s_{i+1}-s_i}, -\frac{2(s_i+s_{i+1})}{s_{i+1}-s_i})$, as well as one data point $\mathbf{q}_i^+ := (\frac{4}{s_{i+1}-s_i}, \frac{2(s_i+s_{i+1})}{s_{i+1}-s_i})$. All these data points receive label $y = 0$.

With the precise description of the selection gadget at hand, we can proceed to proving Lemma 4.

*Proof of Lemma 4.*  First, we focus on the three vertical lines $h_{-\epsilon}$, $h_0$, and $h_\epsilon$. Note each of the three lines contains a sequence of nine data points of which the first three have label 0, the next three have label 1 and the final three have label 0 again. For simplicity, consider one of the three lines and denote these nine data points by $\mathbf{p}_1$ to $\mathbf{p}_9$. Note that $h_0$ contains even more data points, which will become important later. For the following argument, compare Figure 5.

Observe that $f$ restricted to one of the three lines is a one-dimensional, continuous, piecewise linear function with at most four breakpoints. Looking at $\mathbf{p}_2$, $\mathbf{p}_3$, and $\mathbf{p}_4$, the corresponding $y$-labels are 0, 0, and 1, respectively. This can only be fitted if there exists a convex breakpoint between $\mathbf{p}_2$ and $\mathbf{p}_4$. Analogously, there must be a concave breakpoint between $\mathbf{p}_3$ and $\mathbf{p}_5$, another concave breakpoint between $\mathbf{p}_5$ and $\mathbf{p}_7$, and a convex breakpoint between $\mathbf{p}_6$ and $\mathbf{p}_8$. This uses already all four available breakpoints, so there are no other breakpoints. Therefore, the function on the considered line must be linear outside the segment between $\mathbf{p}_2$ and $\mathbf{p}_8$. Since $\mathbf{p}_1$, $\mathbf{p}_2$, $\mathbf{p}_8$, and $\mathbf{p}_9$ all have label 0, it follows

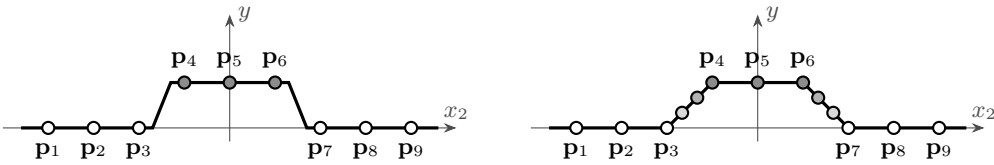

Figure 5: Cross section of the selection gadget through one of the three lines $h_{-\epsilon}$, $h_0$, or $h_\epsilon$. The nine data points (labeled $\mathbf{p}_1$ to $\mathbf{p}_9$) on each of these lines force the function $f$ to attain a "levee-shape" with the exact position and slope of the ascending and descending sections as the only degrees of freedom (left). The four additional data points on $h_0$ even fix these properties and thus exactly determine $f$ on that line (right).

that the function is constant $0$ outside this segment. Moreover, there is no concave breakpoint outside the segment between $\mathbf{p}_3$ and $\mathbf{p}_7$, implying that the function must be convex outside the segment between $\mathbf{p}_3$ and $\mathbf{p}_7$. However, since these two points have label $0$ as well, it follows that $f$ must even be constant $0$ there.

Now consider the segment between $\mathbf{p}_4$ and $\mathbf{p}_6$. There is no convex breakpoint between $\mathbf{p}_4$ and $\mathbf{p}_6$. Therefore, the function must be concave within the segment. Since $\mathbf{p}_4$, $\mathbf{p}_5$, and $\mathbf{p}_6$ all have label $1$, it follows that the function is constant $1$ between $\mathbf{p}_4$ and $\mathbf{p}_6$.

Putting together the insights gained so far, it follows that $f$ restricted to the considered line is constant $0$ first, goes up to constant $1$ via a convex and a concave breakpoint between $\mathbf{p}_3$ and $\mathbf{p}_4$, and goes down to constant $0$ again via a concave and a convex breakpoint between $\mathbf{p}_6$ and $\mathbf{p}_7$ (Figure 5, left). Note that the exact location of these breakpoints and the slope in the sloped segments is not implied by the nine data points considered so far.

This changes, however, when also taking into account the four other data points lying on $h_0$. Combined with the insights so far, they completely determine $f$ on this line (Figure 5, right):

$$f(0, x_2) = \begin{cases} 0, & \text{if } x_2 \leq -2 \text{ or } x_2 \geq 2, \\ 1, & \text{if } -1 \leq x_2 \leq 1, \\ 2 + x_2, & \text{if } -2 \leq x_2 \leq -1, \\ 2 - x_2, & \text{if } 1 \leq x_2 \leq 2. \end{cases}$$

Observe that this is precisely the same as (1) with $x_1 = 0$.

It remains to consider the behavior of $f$ on both sides of $h_0$. To this end, observe that the breakpoints of $f$ restricted to one of the three lines considered so far emerge as intersections of these lines with only four breaklines in total. Let us collect what we know so far about the locations of these four breaklines:

- There are exactly two convex breaklines, intersecting $h_0$ at $(0, -2)$ and $(0, 2)$, respectively. We call them $g_1$ and $g_4$, respectively.

- There are exactly two concave breaklines, intersecting $h_0$ at $(0, -1)$ and $(0, 1)$, respectively. We call them $g_2$ and $g_3$, respectively.

- Each of the four segments

$$I_1 := [(-\epsilon, -2 - \epsilon s_\ell), (-\epsilon, -1 - \epsilon s_1)] \subseteq [(-\epsilon, -7/3), (-\epsilon, -2/3)],$$
$$I_2 := [(-\epsilon, 1 - \epsilon s_\ell), (-\epsilon, 2 - \epsilon s_1)] \subseteq [(-\epsilon, 2/3), (-\epsilon, 7/3)],$$
$$I_3 := [(\epsilon, -2 + \epsilon s_1), (\epsilon, -1 + \epsilon s_\ell)] \subseteq [(\epsilon, -7/3), (\epsilon, -2/3)], \text{ and}$$
$$I_4 := [(\epsilon, 1 + \epsilon s_1), (\epsilon, 2 + \epsilon s_\ell)] \subseteq [(\epsilon, 2/3), (\epsilon, 7/3)]$$

is intersected by exactly one concave and one convex breakline. Here, the inclusions are implied by $\epsilon \leq \min\{\frac{1}{3|s_1|}, \frac{1}{3|s_\ell|}\}$. See Figure 6 for an illustration of the position of these segments.

Now consider $g_2$, which goes through $(0, -1)$, and observe that it cannot intersect $I_2$ for the following reason. If it did, it would intersect $h_\epsilon$ at $x_2 \leq -1 - 5/3 = -8/3 < -7/3$ and hence would neither intersect $I_3$ nor $I_4$. This is a contradiction because there are only two concave breaklines and both $I_3$ and $I_4$ must be intersected by exactly one of them. Consequently, $g_2$ cannot intersect $I_2$, and must intersect $I_1$ instead.

Analogously, it follows that $g_1$ and $g_2$ intersect $I_1$ and $I_3$. Similarly, $g_3$ and $g_4$ intersect $I_2$ and $I_4$. Combining this with the fact that $f$ restricted to each of the three vertical lines $h_{-\epsilon}$, $h_0$, and $h_\epsilon$ has an increasing section from $0$ to $1$ and a decreasing section from $1$ to $0$, this implies that the four lines $g_1$ to $g_4$ do not cross between $h_{-\epsilon}$ and $h_\epsilon$. Let us focus on the quadrilateral enclosed by $g_1$, $g_2$, $h_{-\epsilon}$ and $h_\epsilon$. By what we know so far, $f$ is constant $0$ on $g_1$, constant $1$ on $g_2$, and linear within this quadrilateral. Since $h_{-\epsilon}$ and $h_\epsilon$ are parallel, this implies that the corresponding two sides of the quadrilateral must have the same length. Thus, the quadrilateral must be a parallelogram. In particular, $g_1$ and $g_2$ are parallel. Similarly, $g_3$ and $g_4$ must be parallel.

Let $s$ be the slope of $g_1$ and $g_2$, and let $t$ be the slope of $g_3$ and $g_4$. To complete the proof, we need to show that all four lines are parallel, that is, $s = t$, and that this slope value is equal to $s_i$ for some $i \in [\ell]$.

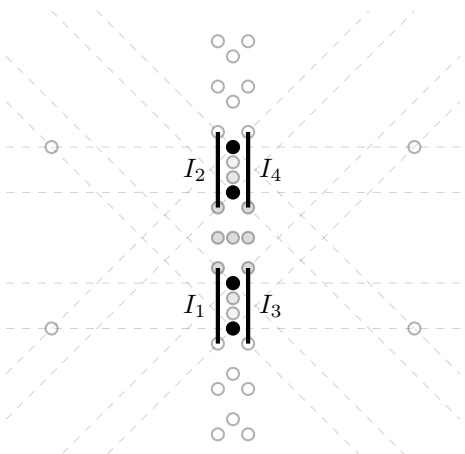

Figure 6: Illustration of the segments $I_1$ to $I_4$ used in the proof of Lemma 4. The figure also highlights (in black) the data points at $(0, -2)$ and $(0, 2)$, each of which lies on a convex breakline, as well as the data points at $(0, -1)$ and $(0, 1)$, each of which lies on a concave breakline.

Without loss of generality, we can assume that $s \leq t$, otherwise we mirror the gadget along the $x_2$-axis. Observe that $s_1 \leq s \leq t \leq s_\ell$ because both $g_1$ and $g_2$ intersect $I_3$, and both $g_3$ and $g_4$ intersect $I_4$.

Let $i^* := \max\{i \mid s_i \leq s\} \in [\ell]$. If $i^* = \ell$, then $s = t = s_\ell$ and we are done. Otherwise, consider the data point $\mathbf{q}_{i^*}^+ = \left( \frac{4}{s_{i^*+1} - s_{i^*}}, \frac{2(s_{i^*} + s_{i^*+1})}{s_{i^*+1} - s_{i^*}} \right)$, which has label $y = 0$.

Let us have a look at what $f$ restricted to the vertical line $h$ through $\mathbf{q}_{i^*}^+$ looks like. By $s \leq t$, the four lines $g_1$, $g_2$, $g_3$, and $g_4$ intersect $h$ in exactly this order (from bottom to top). This means that these lines do not cross between the $x_2$-axis and $h$. By our insights above, this implies that restricted to $h$, $f$ is zero outside the intersection points with $g_1$ and $g_4$, increases from zero to one between $g_1$ and $g_2$, stays constant 1 between $g_2$ and $g_3$, and decreases back to 0 between $g_3$ and $g_4$.

By our choice of $i^*$, we obtain that $s_{i^*+1} > s$. Let us calculate at which $x_2$-coordinate $g_1$ intersects $h$. This happens at

$$x_2 = -2 + \frac{4s}{s_{i^*+1} - s_{i^*}} < -2 + \frac{4s_{i^*+1}}{s_{i^*+1} - s_{i^*}} = \frac{4s_{i^*+1} - 2(s_{i^*+1} - s_{i^*})}{s_{i^*+1} - s_{i^*}} = \frac{2(s_{i^*} + s_{i^*+1})}{s_{i^*+1} - s_{i^*}}.$$

Thus, $\mathbf{q}_{i^*}^+$ lies strictly above the line $g_1$. Since $\mathbf{q}_{i^*}^+$ has label zero, this must imply that $\mathbf{q}_{i^*}^+$ does not lie below $g_4$. Looking at the intersection point of $g_4$ with $h$, this means:

$$2 + \frac{4t}{s_{i^*+1} - s_{i^*}} \leq \frac{2(s_{i^*} + s_{i^*+1})}{s_{i^*+1} - s_{i^*}}$$
$$\Leftrightarrow \quad 2(s_{i^*+1} - s_{i^*}) + 4t \leq 2(s_{i^*} + s_{i^*+1})$$
$$\Leftrightarrow \quad t \leq s_{i^*}.$$

Thus, we obtain $s_{i^*} \leq s \leq t \leq s_{i^*}$, implying that $g_1, g_2, g_3, g_4$ are all parallel and have one of the $\ell$ predefined slopes. This implies that $f$ is the levee $f_{s_{i^*}}$, completing the proof of the lemma. $\qquad \square$

Next, we formalize how to combine multiple selection gadgets. To this end, we define a *selection gadget with offset $z$* as the set of data points of a selection gadget as described above, where we add $z$ to all $x_2$-coordinates of the gadget. In other words, the gadget is centered around the point $(0, z)$.

Now, consider the set of data points originating from $m$ selection gadgets with offsets $z_1, \ldots, z_m$, each one offering the choice between $\ell_j$ many slopes $s_i^{(j)}$, $i \in [\ell_j]$, $j \in [m]$. Suppose further that we uniformly choose $\epsilon := \min_{j \in [m]} \min\left\{ \frac{1}{3|s_1^{(j)}|}, \frac{1}{3|s_\ell^{(j)}|} \right\}$ for all the gadgets such that the vertical lines $h_{-\epsilon}$, $h_0$, and $h_\epsilon$ with $x_1$-coordinates $-\epsilon$, $0$, and $\epsilon$, respectively, each contain either 9 or 13 data

points from each gadget. Let $\delta := \min_{j\in[m]} \min_{i\in[\ell_j-1]}(s_{i+1}^{(j)} - s_i^{(j)})$ be the smallest difference of two consecutive slopes in the $m$ gadgets. Moreover, let $S := \max_{j\in[m]} \max_{i\in[\ell_j]}|s_i^{(j)}|$ be the largest absolute value of all the slopes. In this setting, the following lemma states that fitting all these data points is equivalent to independently choosing one slope for each single gadget and adding up the corresponding levees, provided that the distance of the gadgets is large enough.

**Lemma 11** (Formal version of Lemma 5). *If $z_{j+1} - z_j \geq \frac{8S}{\delta} + 6$ for all $j \in [m-1]$, then there are exactly $\prod_{j=1}^{m} \ell_j$ many continuous piecewise linear functions $f\colon \mathbb{R}^2 \to \mathbb{R}$ with at most $4m$ breaklines fitting the data points of the $m$ selection gadgets, namely $f(x_1, x_2) = \sum_{j=1}^{m} f_{s_{i_j}^{(j)}}(x_1, x_2 - z_j)$ for each choice of indices $i_j \in [\ell_j]$ for each $j \in [m]$.*

*Proof.* We first show that each of these functions does indeed fit all the data points. For this, it is sufficient to show that each levee $f_{s_{i_j}^{(j)}}(x_1, x_2 - z_j)$ is 0 at all the data points $(\bar{x}_1, \bar{x}_2)$ belonging to a selection gadget with index $j' \neq j$. Without loss of generality, we can assume that $z_j = 0$. By the definition of the selection gadget and checking all the possible $x_1$-coordinates, we obtain that $|\bar{x}_1| \leq 4/\delta$. Moreover, looking at the possible $x_2$-coordinates, we obtain that $\bar{x}_2$ can differ at most by $4 + S \cdot |\bar{x}_1|$ from $z_{j'}$, from which we conclude $|\bar{x}_2| \geq |z_{j'}| - 4 - S \cdot |\bar{x}_1| \geq \frac{4S}{\delta} + 2$. On the other hand, all points $(x_1, x_2)$ for which the levee $f_{s_{i_j}^{(j)}}(x_1, x_2)$ is nonzero satisfy $|x_2| < 2 + |s_{i_j} x_1| \leq 2 + S|x_1|$.

Since $|\bar{x}_2| \geq \frac{4S}{\delta} + 2 \geq 2 + S|x_1|$, it follows that $f_{s_{i_j}^{(j)}}$ must be zero at $(\bar{x}_1, \bar{x}_2)$, completing the proof that all claimed functions fit the $m$ selection gadgets.

It remains to show that all functions $f$ fitting the data points of the $m$ selection gadgets are of the claimed form. We show this by induction on $m$. The base case $m = 1$ is given by Lemma 4. Now, let $m \geq 2$ and without loss of generality let $z_1 = 0$. We will again consider the three vertical lines $h_{-\epsilon}$, $h_0$, and $h_\epsilon$ with $x_1$-coordinates $-\epsilon$, 0, and $\epsilon$, respectively. Remember that $f$ restricted to each of these three lines is a one-dimensional continuous piecewise linear function with at most $4m$ breakpoints, stemming from breaklines intersecting the respective vertical line. By looking at each individual gadget and arguing as in the proof of Lemma 4, we obtain the following information:

- There are exactly $2m$ convex breaklines, intersecting $h_0$ at the $2m$ points $(0, z_j - 2)$ and $(0, z_j + 2)$, $j \in [m]$. Note that by our assumptions $z_1 = 0$ and $z_{j+1} - z_j \geq \frac{8S}{\delta} + 6 > 6$, all these points are distinct, two of them are $(0, -2)$ and $(0, 2)$, and all the other $2m - 2$ points lie above the horizontal line $x_2 = 4$.

- There are exactly $2m$ concave breaklines, intersecting $h_0$ at the $2m$ points $(0, z_j - 1)$ and $(0, z_j + 1)$, $j \in [m]$. Again by our assumptions $z_1 = 0$ and $z_{j+1} - z_j \geq \frac{8S}{\delta} + 6 > 6$, all these points are distinct, two of them are $(0, -1)$ and $(0, 1)$, and all the other $2m - 2$ points lie above the horizontal line $x_2 = 5$.

- Each of the four segments $I_1$ to $I_4$ corresponding to the selection gadget with index $j = 1$ as defined in the proof Lemma 4 is intersected by exactly one convex and exactly one concave breakline. There are $4m - 4$ further such segments stemming from selection gadgets with index $j > 1$, and all of those lie completely above the horizontal line $x_2 = 6 - 7/3 = 11/3$.

Looking at the breaklines passing through $(0, -2)$ and $(0, -1)$, they must also pass through one of the described $2m$ segments on $h_{-\epsilon}$ and one of the described $2m$ segments on $h_\epsilon$. Since the considered gadget is the lowest one on the $x_2$-axis, the same argument as in the proof of Lemma 4 applies, which means that the only way of fulfilling these requirements simultaneously is that these breaklines pass through $I_1$ and $I_3$. Once having this, the same argument can be repeated for the breaklines passing through $(0, 1)$ and $(0, 2)$, making use of the fact that all the $4m - 4$ segments not belonging to the considered gadget lie above the $x_2 = 11/3$-line. Therefore, these breaklines must intersect $h_{-\epsilon}$ and $h_\epsilon$ within $I_2$ and $I_4$, respectively.

From this, it follows as in the proof of Lemma 4 that the only way to fit the data points of the selection gadget with index $j = 1$ is one of the $\ell_1$ levees $f_{s_i^{(1)}}$, $i \in [\ell_1]$. Thus, subtracting one of these $\ell_1$ levees from $f$ eliminates four of the $4m$ breaklines. Applying induction to the resulting function and the $m - 1$ remaining selection gadgets completes the proof. $\qquad\square$

Finally, we provide the details of the global layout of our construction. Let $\delta := \frac{1}{2m}$. This will be the smallest difference of any two consecutive slopes in any selection gadget we are going to use. Moreover, no absolute value of a slope will be larger than $S := 1$. From this, we conclude that, in order to apply Lemma 11 in the end, we need to maintain a distance of at least $\Delta := \frac{8S}{\delta} + 6 = 16m + 6$ between the centers of the gadgets.

We start by describing the positions and slopes of the selection gadgets. Compare Figure 4 for an illustration. Firstly, for each clause $C_j$, $j \in [m]$, we introduce one selection gadget with offset $j\Delta$ (that is, centered at $(0, j\Delta)$) and the three different slopes $s_1^{(j)} := (2j-2)\delta - 1$, $s_2^{(j)} := (2j-1)\delta - 1$, and $s_3^{(j)} := 2j\delta - 1$. Note that all these slopes are contained in $[-1, 0]$. The interpretation will be as follows: Choosing the levee with slope $s_r^{(j)}$ for the $j$-th selection gadget corresponds to choosing the $r$-th literal of the $j$-th clause as the one that is set to true. Secondly, for each variable $v_i$, $i \in [n]$, we introduce one selection gadget with offset $-i\Delta$ and the two slopes $-1$ and $1$. Here the interpretation is as follows: choosing the levee with slope $-1$ corresponds to setting the variable to true, while choosing the levee with slope $1$ corresponds to setting the variable to false. Finally, if the $r$-th literal, $r \in [3]$, of clause $C_j$ is $v_i$, then we introduce a data point $\mathbf{p}_{j,r}$ with label $y = 1$ at the intersection of the "center-line" of the levee with slope $s_r^{(j)}$ corresponding to the selection gadget for $C_j$ (that is, the line $x_2 = \Delta j + s_r^{(j)} x_1$) and the "center-line" of the levee with slope $1$ corresponding to the selection gadget of $v_i$ (that is, the line $x_2 = -\Delta i + x_1$). Thus, $\mathbf{p}_{j,r} := (\frac{\Delta(i+j)}{1-s_r^{(j)}}, \frac{\Delta(i+j)}{1-s_r^{(j)}} - \Delta i)$.

This finishes the construction. Before we prove Theorem 1 using this construction, we show the following useful lemma.

**Lemma 12.** *For each $j \in [m]$ and $r \in [3]$, there are exactly two out of the $3m + 2n$ possible levees defined by the selection gadgets which are non-zero at $\mathbf{p}_{j,r}$, namely $f_{s_r^{(j)}}(x_1, x_2 - j\Delta)$ and $f_1(x_1, x_2 + i\Delta)$, where $v_i$ is the $r$-th literal in $C_j$.*

*Proof.* Since $\mathbf{p}_{j,r}$ is the intersection point of the center-lines of the two named levees, it suffices to show that no other levee is non-zero at this point.

Let us start by reminding ourselves that a levee with offset $z$ and slope $s$ is non-zero only for points within a stripe of "vertical width 4", that is, for points $(x_1, x_2)$ with $sx_1 + z - 2 < x_2 < sx_1 + z + 2$.

Now we focus on levees belonging to other clauses $C_{j'}$ with $j' \neq j$. If $j' > j$, then the slope will be at least $s_r^{(j)}$ and the offset will be at least $(j+1)\Delta$. Since $\mathbf{p}_{j,r}$ lies on the right-hand side of the $x_2$-axis and on the center-line of a levee with slope exactly $s_r^{(j)}$ and offset exactly $j\Delta$, we obtain that $\mathbf{p}_{j,r}$ lies below the center-line of the considered levee with a vertical distance of at least $\Delta > 2$, implying that the levee must vanish at $\mathbf{p}_{j,r}$. In the case $j' < j$ it follows similarly with $\mathbf{p}_{j,r}$ lying above instead of below the considered levee.

Next, let us focus on the two levees belonging to the same clause $C_j$ but to the $r'$-th literal with $r' \neq r$. The slope of such a levee differs by at least $\delta$ from $s_r^{(j)}$, while the offset is exactly $j\Delta$. This implies that $\mathbf{p}_{j,r}$ has a vertical distance of at least $\delta \frac{\Delta(i+j)}{1-s_r^{(j)}} \geq \delta \frac{2\Delta}{2} = \delta\Delta > 8 > 2$ from the center-line of the considered levee.

Next, let us focus on a levee with slope $1$ belonging to a variable $v_{i'}$ with $i' \neq i$. Since $\mathbf{p}_{j,r}$ lies on the center-line of the levee with slope $1$ belonging to $v_i$, these levees are parallel, and have vertical distance at least $\Delta > 2$, this case is settled, too.

Finally, let us focus on a levee with slope $-1$ belonging to any variable. Such a levee has an offset of at most $-\Delta$ and its slope is at most $s_r^{(j)}$. Since $\mathbf{p}_{j,r}$ lies on the center-line of the levee with offset $j\Delta$ and slope $s_r^{(j)}$, this implies that its vertical distance to the considered levee is at least $2\Delta > 2$, finishing the proof. □

Finally, we are ready to prove the main theorem.

*Proof of Theorem 1.* We reduce from POITS and construct an instance of 2L-RELU-NN-TRAIN($\mathcal{L}$) with $k = 4(m + n)$ and $\gamma = 0$ as described above. Note that, overall, we introduce $O(m + n)$ points with rational coordinates (with $\text{poly}(m, n)$ bits) which are polynomial-time computable.

To prove equivalence between the POITS instance and the constructed instance, let us first assume that the POITS instance is a yes-instance. Let $T \subseteq [n]$ be a set of indices such that the truth assignment with $v_i = $ true for $i \in T$ and $v_i = $ false for $i \notin T$ sets exactly one literal per clause to true. Let $r_j \in \{1, 2, 3\}$ denote which of the three literals is set to true in clause $C_j$ by this assignment. We claim that the following function, which is a sum of $m + n$ levees and thus realizable with $k = 4(m + n)$ ReLUs using Observation 3, exactly fits all the constructed data points:

$$f(x_1, x_2) = \sum_{i \in T} f_{-1}(x_1, x_2 + i\Delta) + \sum_{i \notin T} f_1(x_1, x_2 + i\Delta) + \sum_{j=1}^{m} f_{s_{r_j}^{(j)}}(x_1, x_2 - j\Delta). \quad (2)$$

By Lemma 11, $f$ fits all data points belonging to the selection gadgets. It remains to show that $f$ attains value 1 at all the data points $\mathbf{p}_{j,r}$, $j \in [m]$, $r \in [3]$. To see this, fix such $j$ and $r$ and let the $r$-th literal in $C_j$ be $v_i$. By Lemma 12, the only two levees which can potentially be non-zero at $\mathbf{p}_{j,r}$ are $f_{s_r^{(j)}}(x_1, x_2 - j\Delta)$ and $f_1(x_1, x_2 + i\Delta)$. If $r = r_j$, then $v_i = $ true and the former levee attains value 1 while the latter levee attains value 0 at $\mathbf{p}_{j,r}$. Otherwise, if $r \neq r_j$, then $v_i = $ false and the former levee attains value 0 while the latter levee attains value 1 at $\mathbf{p}_{j,r}$. In both cases, the data point is fitted correctly.

Now suppose conversely that the constructed data points can be precisely fitted with a function $f$ representable with $k = 4(m + n)$ ReLUs. By Lemma 11, $f$ must be of the form (2) for some set $T \subseteq [n]$ and some values $r_j \in [3]$ for all $j \in [m]$. We claim that setting $v_i = $ true for $i \in T$ and $v_i = $ false for $i \notin T$ sets exactly one literal per clause to true. To see this, fix $j \in [m]$ and $r \in [3]$ and let $v_i$ be the $r$-th literal of $C_j$. Using Lemma 12 again, observe that exactly one of the two levees $f_{s_r^{(j)}}(x_1, x_2 - j\Delta)$ and $f_1(x_1, x_2 + i\Delta)$ must belong to the sum (2) because the data point $\mathbf{p}_{j,r}$ has label one. In other words, it holds that either $r = r_j$ (implying $i \in T$) or $i \notin T$. This implies that, for each $j \in [m]$, the defined truth assignment sets exactly the $r_j$-th literal of $C_j$ to true, finishing the overall proof. □

## B   Detailed Proof of W[1]-Hardness for Four ReLUs

We prove Theorem 6 with a parameterized reduction from 2-HYPERPLANE SEPARABILITY. The hardness proof for 2-HYPERPLANE SEPARABILITY by Giannopoulos et al. [2009] in fact shows that if there is a solution, then there is a solution where $Q$ lies entirely in one region of the hyperplane arrangement and the points in $P$ lie only in the two neighboring regions. Formally, if the two hyperplanes are defined by $\mathbf{h}_i \cdot \mathbf{x} + o_i = 0$ for $\mathbf{h}_i \in \mathbb{R}^d$, $o_i \in \mathbb{R}$, $i \in [2]$, then (without loss of generality) we can assume that the following holds:

$$\forall \mathbf{q} \in Q : \mathbf{h}_1 \cdot \mathbf{q} + o_1 > 0 > \mathbf{h}_2 \cdot \mathbf{q} + o_2 \quad (3)$$
$$\forall \mathbf{p} \in P : \text{sgn}(\mathbf{h}_1 \cdot \mathbf{p} + o_1) = \text{sgn}(\mathbf{h}_2 \cdot \mathbf{p} + o_2) \quad (4)$$

Moreover, a closer inspection of their reduction shows that one can assume that the hyperplanes have distance at least $\epsilon := m^{-3}$ to each input point[3]. That is, we can assume

$$\forall \mathbf{x} \in Q \cup P, i \in [2] : \frac{|\mathbf{h}_i \cdot \mathbf{x} + o_i|}{\|\mathbf{h}_i\|} > \epsilon. \quad (5)$$

We will make use of these assumptions in the following proof.

*Proof of Theorem 6.* Let $(Q, P)$ be an instance of (restricted) 2-HYPERPLANE SEPARABILITY and let $m := |Q \cup P|$ and $\epsilon := m^{-3}$. We construct the instance $(X \subseteq \mathbb{R}^{d+1}, k := 4, \gamma := 0)$ of 2L-RELU-NN-TRAIN($\mathcal{L}$), where $X$ contains the following points:

- $(\mathbf{q}, 1)$ for each $\mathbf{q} \in Q$,

- $(\mathbf{p}, 0)$ for each $\mathbf{p} \in P$,

---

[3]The critical points in the reduction are the constraint points $q_{ij}^{uv}$ which are separated from the points $p_{iu_i}, p_{i\overline{u}_i}, p_{ju_j}, p_{j\overline{u}_j}$ by some translation of the hyperplane $H(u_1, \ldots, u_k)$ towards the origin. The distance of any $q_{ij}^{uv}$ to $H(u_1, \ldots, u_k)$ is at least $2\sin^3(\pi/m) \geq 2m^{-3}$ in one dimension.

- $(\mathbf{r_{qp}} := (1-\delta)\mathbf{q} + \delta\mathbf{p}, 1)$ and $(\mathbf{s_{qp}} := \delta\mathbf{q} + (1-\delta)\mathbf{p}, 0)$ for each $(\mathbf{q}, \mathbf{p}) \in Q \times P$, where $\delta := \epsilon(2\|\mathbf{q} - \mathbf{p}\|)^{-1}$.

Note that $\mathbf{r_{qp}}$ $(\mathbf{s_{qp}})$ lies on the line segment $\mathbf{qp}$ at distance $\epsilon/2$ to $\mathbf{q}$ $(\mathbf{p})$. Overall, we construct $n := |X| \in O(m^2)$ points, which can be done in polynomial time.

For the correctness, assume first that there are two hyperplanes $\mathcal{H}_i$, $i \in [2]$, defined by $\mathbf{h}_i \cdot \mathbf{x} + o_i = 0$ (wlog $\|\mathbf{h}_i\| = 1$) that strictly separate $Q$ and $P$ and satisfy (3)–(5).

A solution for $(X, 4, 0)$ can then be constructed as follows (see also Figure 7): We use two ReLUs realizing an "upward step" of height 1 (with slope $\beta := 4/\epsilon$) in the direction of $\mathbf{h}_1$. That is, we set

$$\mathbf{w}_1 := \beta\mathbf{h}_1, \qquad b_1 := \beta o_1, \qquad a_1 := 1,$$
$$\mathbf{w}_2 := \beta\mathbf{h}_1, \qquad b_2 := \beta o_1 - 1, \qquad a_2 := -1.$$

Additionally, we use two ReLUs realizing a "downward step" of height 1 (with slope $-\beta$) in the direction of $\mathbf{h}_2$, that is,

$$\mathbf{w}_3 := \beta\mathbf{h}_2, \qquad b_3 := \beta o_2, \qquad a_3 := -1,$$
$$\mathbf{w}_4 := \beta\mathbf{h}_2, \qquad b_4 := \beta o_2 - 1, \qquad a_4 := 1.$$

Let $\mathcal{W}_i$ be the hyperplane defined by $\mathbf{w}_i \cdot \mathbf{x} + b_i = 0$ for $i \in [4]$. Note that $\mathcal{W}_1 = \mathcal{H}_1$ and $\mathcal{W}_3 = \mathcal{H}_2$. Note further that $\mathcal{W}_2$ is parallel to $\mathcal{W}_1$ at distance $\beta^{-1} = \epsilon/4$ and $\mathcal{W}_4$ is parallel to $\mathcal{W}_3$ at distance $\epsilon/4$.

To verify that all data points are exactly fitted, consider first a point $\mathbf{q} \in Q$. From (3) and (5), we obtain

$$\mathbf{w_1} \cdot \mathbf{q} + b_1 = \beta(\mathbf{h}_1 \cdot \mathbf{q} + o_1) > 0,$$
$$\mathbf{w_2} \cdot \mathbf{q} + b_2 = \beta(\mathbf{h}_1 \cdot \mathbf{q} + o_1 - \beta^{-1}) > \beta\epsilon - 1 > 0,$$
$$\mathbf{w_3} \cdot \mathbf{q} + b_3 = \beta(\mathbf{h}_2 \cdot \mathbf{q} + o_2) < 0,$$
$$\mathbf{w_4} \cdot \mathbf{q} + b_4 = \beta(\mathbf{h}_2 \cdot \mathbf{q} + o_2 - \beta^{-1}) < 0.$$

From the above inequalities, it follows

$$\phi(\mathbf{q}) = \beta(\mathbf{h}_1 \cdot \mathbf{q} + o_1) - \beta(\mathbf{h}_1 \cdot \mathbf{q} + o_1 - \beta^{-1}) = 1.$$

Now consider a point $\mathbf{r_{qp}}$ and note that, for each $\mathcal{W}_i$, $\mathbf{r_{qp}}$ lies in the same half-space as $\mathbf{q}$ since it has distance $\epsilon/2$ to $\mathbf{q}$ which has distance at least $\frac{3}{4}\epsilon$ to $\mathcal{W}_i$ (by (5)). Thus,

$$\phi(\mathbf{r_{qp}}) = \beta(\mathbf{h}_1 \cdot \mathbf{r_{qp}} + o_1) - \beta(\mathbf{h}_1 \cdot \mathbf{r_{qp}} + o_1 - \beta^{-1}) = 1.$$

Next, consider a point $\mathbf{p} \in P$. Using (4) and (5), one easily verifies that

$$\mathrm{sgn}(\mathbf{w_1} \cdot \mathbf{p} + b_1) = \mathrm{sgn}(\mathbf{w_2} \cdot \mathbf{p} + b_2) = \mathrm{sgn}(\mathbf{w_3} \cdot \mathbf{p} + b_3) = \mathrm{sgn}(\mathbf{w_4} \cdot \mathbf{p} + b_4).$$

Hence, $\phi(\mathbf{p}) = 0$ clearly holds if all the above signs are negative. If all signs are positive, then

$$\phi(\mathbf{p}) = \beta(\mathbf{h}_1 \cdot \mathbf{p} + o_1) - \beta(\mathbf{h}_1 \cdot \mathbf{p} + o_1 - \beta^{-1}) - \beta(\mathbf{h}_2 \cdot \mathbf{p} + o_2) + \beta(\mathbf{h}_2 \cdot \mathbf{p} + o_2 - \beta^{-1}) = 0.$$

Finally, any point $\mathbf{s_{qp}}$ analogously lies in the same half-space as $\mathbf{p}$ for each $\mathcal{W}_i$, which also implies $\phi(\mathbf{s_{qp}}) = 0$. Thus, all points are correctly fitted.

Conversely, assume that the points in $X$ can be exactly fitted by $\phi$ realized by four ReLUs with values $\mathbf{w}_i, b_i, a_i, i \in [4]$. Let $I^+ := \{i \in [4] \mid a_i = 1\}$ and $I^- := \{i \in [4] \mid a_i = -1\}$.

Consider an arbitrary line segment $\mathbf{qp}$ for $(\mathbf{q}, \mathbf{p}) \in Q \times P$. Clearly, the points $(\mathbf{q}, 1)$, $(\mathbf{r_{qp}}, 1)$ and $(\mathbf{s_{qp}}, 0)$ on this line segment cannot all lie on the same piece of $\phi$. Hence, $\phi$ must have a concave breakpoint at some point on the open segment between $\mathbf{q}$ and $\mathbf{p}$. That is, there must be a ReLU $i \in I^-$ such that the hyperplane defined by $(\mathbf{w}_i, b_i)$ intersects the open line segment $\mathbf{qp}$ and does not contain $\mathbf{q}$ or $\mathbf{p}$. Analogously, the points $(\mathbf{p}, 0)$, $(\mathbf{s_{qp}}, 0)$ and $(\mathbf{r_{qp}}, 1)$ enforce a convex breakpoint, that is, a ReLU $j \in I^+$ with a hyperplane $(\mathbf{w}_j, b_j)$ also intersecting the open line segment $\mathbf{qp}$ and not containing $\mathbf{q}$ or $\mathbf{p}$.

To sum up, every open line segment $\mathbf{qp}$ is intersected by at least two hyperplanes (not containing $\mathbf{q}$ or $\mathbf{p}$), one corresponding to a ReLU $i \in I^{-1}$ and one corresponding to a ReLU $j \in I^+$. Since

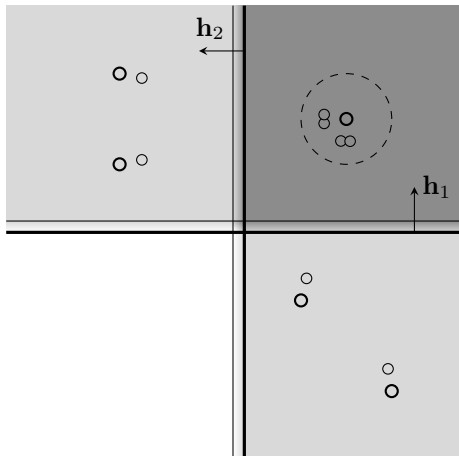

Figure 7: Example of the reduction from 2-HYPERPLANE SEPARABILITY for $d = 2$ dimensions. Big points are points in $Q$ (dark gray) and in $P$ (light gray). The small points are additionally introduced. The four lines are the breaklines of the four ReLUs. The two thick lines indicate the original two separating lines. The dashed circle has radius $\epsilon$.

there are only four ReLUs, it follows that $\min(|I^+|, |I^-|) \leq 2$. That is, we obtain a solution for 2-HYPERPLANE SEPARABILITY by picking either all hyperplanes corresponding to $I^+$ or all hyperplanes corresponding to $I^-$.

This finishes the reduction. Note that since the dimension of the input data points in our constructed instance is $d$, any algorithm solving 2L-RELU-NN-TRAIN($\mathcal{L}$) in time $\rho(d)n^{o(d)}\operatorname{poly}(L)$ would imply an algorithm running in time $\rho(d)m^{o(d)}\operatorname{poly}(L')$ for 2-HYPERPLANE SEPARABILITY contradicting the ETH. □

## C    Proof of Fixed-Parameter Tractability for the Convex Case

Before proving Theorem 10, we introduce some definitions. For $I \subseteq [k]$, let $R_I \subseteq \mathbb{R}^d$ be the *active region* of the ReLUs in $I$, that is, $\mathbf{x} \in R_I$ if and only if

$$\forall j \in I : \mathbf{w}_j \mathbf{x} + b_j \geq 0,$$
$$\forall j \in [k] \setminus I : \mathbf{w}_j \mathbf{x} + b_j \leq 0.$$

Note that $R_I$ could be empty. Clearly, on each $R_I$, $\phi$ is the affine function $\sum_{j \in I}(\mathbf{w}_j \cdot \mathbf{x} + b_j)$. Let $F_I := \{(\mathbf{x}, \phi(\mathbf{x})) \mid \mathbf{x} \in R_I)\}$ be the piece corresponding to $I$.

The convexity of $\phi$ now allows for a branching algorithm assigning the input data points to the at most $2^k$ pieces.

*Proof of Theorem 10.* Let $(\mathbf{x}_1, y_1), \ldots, (\mathbf{x}_n, y_n) \in \mathbb{R}^{d+1}$, $k \in \mathbb{N}$, and let $L$ denote the overall number of input bits. The idea is to use a search tree algorithm to check whether the data can be exactly fitted with $k$ (convex) ReLUs. To this end, we define $2^k$ sets $S_1, \ldots, S_{2^k}$ where each $i \in [2^k]$ one-to-one corresponds to a certain subset $I(i) \subseteq [k]$ of active ReLUs. For given point sets $S \subseteq \mathbb{R}^{d+1}$ and $S_i \subseteq \mathbb{R}^{d+1}$, $i \in [2^k]$, our algorithm checks whether the points in $S$ can be exactly fitted by $k$ ReLUs with the additional constraint that $S_i \subseteq F_{I(i)}$ holds for each $i \in [2^k]$. That is, the following (in)equalities must hold

$$\mathbf{x} \in R_{I(i)} \text{ and } \sum_{j \in I(i)} \mathbf{w}_j \mathbf{x} + b_j = y, \quad i \in [2^k], (\mathbf{x}, y) \in S_i. \tag{6}$$

Algorithm 1 depicts the pseudocode of our `ExactFit` algorithm. We solve an instance with an initial call where $S := \{(\mathbf{x}_1, y_1), \ldots, (\mathbf{x}_n, y_n)\}, S_1 = S_2 = \cdots = S_{2^k} := \emptyset$.

---

**Algorithm 1:** ExactFit$(S, S_1, \ldots, S_{2^k})$

---

**1** **if** $S = \emptyset$ **then**
**2** | **return** check-feasibility$(S_1, \ldots, S_{2^k})$
**3** **else**
**4** | choose $(\mathbf{x}, y) \in S$
**5** | $S \leftarrow S \setminus \{(\mathbf{x}, y)\}$
**6** | **foreach** $i = 1, \ldots, 2^k$ **do**
**7** | | $S_i \leftarrow S_i \cup \{(\mathbf{x}, y)\}$
**8** | | check-forced-points$(S, S_1, \ldots, S_{2^k})$
**9** | | $a \leftarrow$ ExactFit$(S, S_1, \ldots, S_{2^k})$
**10** | | **if** $a =$ Yes **then return** Yes
**11** **return** No

---

**Algorithm 2:** check-forced-points$(S, S_1, \ldots, S_{2^k})$

---

**1** **foreach** $(\mathbf{x}, y) \in S$ **do**
**2** | **foreach** $i = 1, \ldots, 2^k$ **do**
**3** | | $\mu \leftarrow$ lower-bound$(\mathbf{x}, i, S_1, \ldots, S_{2^k})$
**4** | | **if** $\mu = y$ **then**
**5** | | | $S_i \leftarrow S_i \cup \{(\mathbf{x}, y)\}$
**6** | | | $S \leftarrow S \setminus \{(\mathbf{x}, y)\}$
**7** | | | **restart**
**8** | | **if** $\mu > y$ **then**
**9** | | | **reject branch** (**return** No)

---

The correctness of Algorithm 1 follows by induction on $|S|$. For $S = \emptyset$, we simply need to check whether the system (6) of linear (in)equalities is feasible. This can be done by solving a linear program with $k(d + 1)$ variables and $O(n)$ constraints in $O(\text{poly}(k, L))$ time (this is done by check-feasibility in Line 2).

If $S \neq \emptyset$ and $(S, S_1, \ldots, S_{2^k})$ is a no-instance, then none of the recursive calls in Line 9 will be successful (by induction). Hence, the algorithm correctly returns "No" in Line 11.

Now assume that $(S, S_1, \ldots, S_{2^k})$ is a yes-instance. Then, any point $(\mathbf{x}, y) \in S$ must lie on some piece $F_{I(i)}$. That is, $(\mathbf{x}, y)$ can be put into some $S_i$. Hence, in Line 6, we branch into all $2^k$ options. In each branch, we then check whether putting $(\mathbf{x}, y)$ into $S_i$ also forces other points from $S$ (due to assumed convexity) to be contained in some $S_{i'}$ (this is done by check-forced-points in Line 8). We do this in order to achieve our claimed running time bound as we will show later.

The pseudocode for this check is given in Algorithm 2. The idea is to compute for each $(\mathbf{x}, y) \in S$ and each $i \in [2^k]$ the lower bound

$$\mu := \min_{\mathbf{w}_j, b_j} \sum_{j \in I(i)} (\mathbf{w}_j \mathbf{x} + b_j)$$

subject to the constraints (6), which again can be accomplished via linear programming in $O(\text{poly}(k, L))$ time. Note that both $\mu = +\infty$ (linear program is infeasible) and $\mu = -\infty$ (linear program is unbounded) are possible. This is done by lower-bound in Line 3. Now, note that $\mu > y$ implies that

$$\phi(\mathbf{x}) = \sum_{j=1}^{k} [\mathbf{w}_j \mathbf{x} + b_j]_+ \geq \sum_{j \in I(i)} (\mathbf{w}_j \mathbf{x} + b_j) > y$$

holds for every $\phi$ satisfying (6). That is, we can reject (Line 9) the current branch of ExactFit. If $\mu = y$, then we have

$$\phi(\mathbf{x}) \geq \sum_{j \in I(i)} (\mathbf{w}_j \mathbf{x} + b_j) = y$$

for every $\phi$ satisfying (6), and thus we can safely put $(\mathbf{x}, y)$ into $S_i$. To see that this is correct, assume that a solution puts $(\mathbf{x}, y) \in F_{I'}$ for some $I' \subseteq [k]$ with $I' \neq I(i)$. Then, we have

$$
\begin{aligned}
y &= \sum_{j \in I'} (\mathbf{w}_j \mathbf{x} + b_j) = \sum_{j \in I' \cap I(i)} (\mathbf{w}_j \mathbf{x} + b_j) + \sum_{j \in I' \setminus I(i)} (\mathbf{w}_j \mathbf{x} + b_j) \\
&= \sum_{j \in I(i)} (\mathbf{w}_j \mathbf{x} + b_j) = \sum_{j \in I' \cap I(i)} (\mathbf{w}_j \mathbf{x} + b_j) + \sum_{j \in I(i) \setminus I'} (\mathbf{w}_j \mathbf{x} + b_j),
\end{aligned}
$$

which implies

$$
\sum_{j \in I' \setminus I(i)} (\mathbf{w}_j \mathbf{x} + b_j) = \sum_{j \in I(i) \setminus I'} (\mathbf{w}_j \mathbf{x} + b_j).
$$

Since $\mathbf{x} \in R_{I'}$, it follows that the left sum is at least zero and the right sum is at most zero. Thus, both sums are zero and $\mathbf{w}_j \mathbf{x} + b_j = 0$ holds for all $j \in (I' \setminus I(i)) \cup (I(i) \setminus I')$, which shows that $\mathbf{x} \in R_{I(i)}$. Thus, putting $(\mathbf{x}, y)$ into $S_i$ is correct.

Note that adding a point to $S_i$ adds new constraints to (6). Hence, we restart the procedure (Line 7) to check whether this forces new points. Overall, `check-forced-points` takes $O(n^2 2^k \operatorname{poly}(k, L)) \subseteq O(2^k \operatorname{poly}(k, L))$ time.

As regards the correctness of Algorithm 1 now, note that `check-forced-points` clearly never incorrectly rejects a branch of `ExactFit` and never forces points incorrectly. Hence, one of the recursive calls in Line 9 will correctly answer "Yes" (by induction), which proves the correctness.

It remains to analyze the running time of Algorithm 1. Clearly, each call to the algorithm takes $O(2^k \operatorname{poly}(k, L))$ time and recursively branches into $2^k$ options. It remains to bound the depth of the recursion tree. To this end, note that the recursion stops as soon as $S$ is empty or the current branch is rejected by Algorithm 2. We claim that the latter happens after at most $k(d+1) + 1$ recursive calls.

To verify this claim, observe that the algorithm maintains the invariant that the linear program

$$
\min_{\mathbf{w}_j, b_j} \sum_{j \in I(i)} (\mathbf{w}_j \mathbf{x} + b_j) \quad \text{s.t. (6)}
$$

has a solution $\mu < y$ for every $i \in [2^k]$ and $(\mathbf{x}, y) \in S$. This invariant is achieved by checking for forced points in Line 8. Let $P \subseteq \mathbb{R}^{k(d+1)}$ be the polyhedron defined by (6) in the variables $(\mathbf{w}_j, b_j)_{j=1,\ldots,k}$. Now, adding a point $(\mathbf{x}, y)$ to some $S_i$ (Line 7) adds constraints to (6) which yield a polyhedron contained in

$$
P' := P \cap \{ (\mathbf{w}_j, b_j)_{j=1,\ldots,k} \mid \sum_{j \in I(i)} (\mathbf{w}_j \mathbf{x} + b_j) = y \}.
$$

By the above invariant, there exists a $(\mathbf{w}_j, b_j)_{j=1,\ldots,k} \in P$ with $\sum_{j \in I(i)} (\mathbf{w}_j \mathbf{x} + b_j) < y$. Hence, $\operatorname{aff}(P') \subsetneq \operatorname{aff}(P)$ and $\dim(P') < \dim(P)$. That is, each recursive call decreases the dimension of the feasible polyhedron. Thus, after at most $k(d+1) + 1$ recursive calls we reach an empty polyhedron, in which case the current branch is rejected.

To sum up, we obtain an overall running time of $2^{O(k^2 d)} \operatorname{poly}(k, L)$. $\qquad \square$

Note that if positive and negative coefficients $a_j$ are allowed, then our search tree approach of Algorithm 1 does not work since we cannot check for forced points anymore which is necessary to ensure a bounded recursion depth. Indeed, Theorem 6 implies that this approach cannot work already for $k = 4$. It is unclear whether this issue can be resolved for $k = 2$ or $k = 3$.