# OpenReview forum: "Training Neural Networks is NP-Hard in Fixed Dimension"
_NeurIPS.cc/2023/Conference — NeurIPS 2023 poster_

### Official Review · Reviewer_2aQG · 2023-07-05

**Soundness:** 3 good
**Presentation:** 3 good
**Contribution:** 3 good
**Rating:** 6
**Confidence:** 4

**Summary:**

The authors analyze the fpt of training NN. Their main result is that finding a global minimum is hard even for fixed dimension.

**Strengths:**

I like the direction of studying the FPT of training NN. The reductions are new to the best of my knowledge and could find further applications.


**Weaknesses:**

The authors mention generalization a few times. However, the setting considered is completely detached from generalization.
For fixed dimension the sample complexity of the learning problem is fixed and hence can be solved trivially. A more general concern is that low dimension is a rarity in ML applications. In this regard, the number of neurons k seems to be much more interesting.

"We assume basic knowledge on computational complexity theory." I think this sentence can safely be removed.

There is a large literature on what can and cannot be done efficiently with respect to learning thresholds that the authors ignore.
One example is:
"Adam Klivans and Pravesh Kothari. Embedding hard learning problems into gaussian space" and various other papers (regarding learning ReLUs) by Kane and Diakonikolas. While the hardness in these is not NP-hardness, they should be mentioned nevertheless.

**Questions:**

What is the best running-time dependency in term of k when can expect for fixed dimension? What is the running time of the fastest exponential algorithm for learning shallow ReLUs? Thresholds? Are their improvements for the running time of the algorithm of the Arora et al., paper?



**Limitations:**

Proving hardness results for low dimension.

---

> ### Author Rebuttal · Authors · 2023-08-03
>
> Thank you very much for your review.
>
> Indeed our paper is not concerned about generalization.
> The goal is to understand the complexity of training in fixed dimension (which was an open question).
> Note that hardness for few dimensions implies hardness for arbitrary dimensions (as we explained) and is thus a stronger result.
> As regards the number $k$ of hidden neurons, hardness is already known for a single neuron.
> We additionally show W[1]-hardness w.r.t. $d$ for 4 hidden neurons with zero training error.
>
> Thank you for pointing out the papers from the literature.
> We consider adding them to our references for the final version.
>
> As regards your questions, note that our hardness results prove that the algorithm by Arora et al. is essentially optimal, that is, $n^{dk}$ is best possible in general (also for linear threshold activation). However, for the convex case, we give an improved algorithm running in $2^{O(k^2d)}$ time.
> The best running time regarding $k$ for fixed $d$ is still open (Question 2). A running time of $f(k)\textrm{poly}(n)$ might be possible. This is our major open question.

---

> > ### Comment · Reviewer_2aQG · 2023-08-13
> > **Response**
> >
> > Wasn't the hardness for non constant d already known?
> >
> > I encourage the authors to add a table about known running times for the various problems they consider as well as mention the open problem stated in the response to this review.
> >
> > I keep my score.

---

> > > ### Author Response · Authors · 2023-08-17
> > >
> > > Thank you for your follow-up questions.
> > >
> > > Yes, hardness for non-constant $d$ was already known before. We cited this in related work.
> > >
> > > We described the known running times in detail in the Introduction section. We will consider to add a table.
> > >
> > > Note that the open problem is explicitly stated in the conclusion already.

---

### Official Review · Reviewer_bRY2 · 2023-07-07

**Soundness:** 3 good
**Presentation:** 4 excellent
**Contribution:** 3 good
**Rating:** 8
**Confidence:** 2

**Summary:**

The authors show that learning a 2-layered neural network with ReLU activation function, in an underparameterized regime, is NP-hard when the training error is 0. Specifically, they highlight that such learning cannot be accomplished in time complexity dependent on $k^{f(d)}$, where $k$ is the number of neurons in the layer and $d$ is the input dimension. Their main technique is a reduction from the NP-complete problem of POSITIVE ONE-IN-THREE SAT. In their reduction, they strategically position selection gadgets (corresponding to clauses, variables in POITS, and additional data points) along the y-axis, ensuring sufficient spacing between them. This construction can be represented by a piecewise linear function or a combination of ReLUs if and only if it implies a true instance of POITS.

They also show W[1]-hardness with respect to $d$ for $k = 4$ by reducing the problem to 2 HYPERPLANE-SEPARABILITY. This problem determines whether two given point sets can be strictly separated by two hyperplanes. Additionally, they prove that linear thresholding for 2-layered neural networks is also NP-hard, with a similar reduction as proposed for ReLU activations. Finally, they propose a branching algorithm for learning a 2-layered neural network, which has exponential time complexity with respect to $k$ and $d$.

**Strengths:**

The authors have effectively motivated the problem and provided a good background for the problem. The part where they describe the geometry of $\phi$ and define the concept of a levee, accompanied by illustrative figures, was particularly informative and interesting.

**Weaknesses:**

The variable $\ell$ seems to be overloaded, as it is used for both the loss function and the number of possible levees. This can be confusing when it is introduced again in the selection gadget.

**Questions:**

* When you mention "full dimensional cells," are you referring to cells of dimension $d$?
* In order to prevent the other hyperplanes from becoming convex or concave, does the breakpoint $\mathbf{x}$ need to lie exclusively on one hyperplane?

**Limitations:**

The authors discuss various approaches to extend their construction to obtain similar results in higher input dimensions beyond 2, for a fixed training error $\ge 0$ and for more than 4 ReLUs for the W[1] hardness result. Additionally, they address the possibilities of extending to piecewise linear ReLU and piecewise constant linear thresholding activations and also acknowledge that their techniques might not be applicable to smooth activations.

Finally they address two interesting open questions: 1. Results for the task of minimising generalization error instead of training error and 2. Training neural networks for approximate optimality.

---

> ### Author Rebuttal · Authors · 2023-08-03
>
> Thank you very much for your review.
>
> We will fix the issue with the variable $\ell$.
>
> Yes, "full-dimensional" means $d$-dimensional. We will make this precise.
>
> Yes, a breakpoint lies on only one hyperplane. We will fix this.

---

### Official Review · Reviewer_pi8Y · 2023-07-07

**Soundness:** 4 excellent
**Presentation:** 3 good
**Contribution:** 2 fair
**Rating:** 6
**Confidence:** 3

**Summary:**

This paper mainly studies the complexity of training two-layer ReLU networks when they are not over-parameterized. Specifically, the authors prove that training a two-layer ReLU network to zero or arbitrary loss value is NP-hard when the input dimension $d\geq 2$. This result answers an open question (Question 1) posed by Arora et al. (2018) negatively by proving the NP-hardness. On the other hand, by assuming the exponential time hypothesis, the paper also partially answers a more general question of whether a running time that is polynomial in the data size or the number of hidden nodes (Question 2). The paper also provides a positive answer to Question 2 when the ReLU network is assumed to compute a convex function.

**Strengths:**

The novelty of the paper is clear. It provides answers to open questions on the complexity of training two-layer ReLU networks. Therefore, the results seem to be significant. The presentation is concise and easy to follow. I enjoy reading the paper.

**Weaknesses:**

Usually, there are symmetries and redundancies in the dataset. It would be more convincing and realistic if the authors could give some insights on how to leverage these available properties to develop efficient training algorithms. In practice, training a two-layer ReLU network is very efficient. Hence, I feel the settings in this paper seem to be a bit unrealistic. My concern is that it seems hard to apply these results to make a positive impact on our machine learning community. One possible way to address this is to discuss assumptions on different aspects (e.g., architectures) to make training tractable or even efficient. This work focuses on two-layer ReLU networks, but it would be interesting to give insights into extending the results to networks with multiple layers.

**Questions:**

1. Line 35: In the case when $\gamma$ is strictly positive, are there any extra assumptions imposed on the loss function? The only assumption on the loss function $\ell(x,y)=0$ iff $x=y$ is fairly general. It would be more convincing if the authors could clarify more about the loss function.

2. Would it yield the same conclusion as Theorem 1 if a regularizer (e.g., weight decay or L1) is added to the loss function ($k<n$)?

3. Would it be possible to consider some assumptions on architectures that give a polynomial-time algorithm for training a two-layer ReLU network?

**Limitations:**

The authors have addressed some of the limitations of this work in the text. Other limitations in my view are the gaps between the settings of the problem and the training of ReLU networks in practice. For example, one may be able to make some assumptions on architectures, regularizations, and properties of data to derive a polynomial-time training algorithm. However, given that the main purpose of this paper is to answer the open questions, discussions on these aspects may be skipped or minimized.

---

> ### Author Rebuttal · Authors · 2023-08-03
>
> Thank you very much for your review.
>
> Our goal was to settle the complexity of the training problem in the general case (without further assumptions on the input), which turned out to be hard. We agree with you that this is a worst-case result, which does not necessarily reflect practical conditions, but we are convinced that such results belong to an important fundamental understanding of the training problem.
> Indeed our hardness results hold for a very simple restricted setting which makes them even stronger (e.g. for more than two layers, the problem can be expected to be at least as hard).
> As regards impact, our results yield a better theoretical understanding of the power and expressiveness of ReLU networks and settle the complexity status almost completely.
> Moreover, our results can be seen as a justification for making certain assumptions on the input data in order to achieve polynomial time (since without any assumptions, polynomial time is provably unlikely).
>
> Adressing your questions:
>
> 1. There are no further assumptions on the loss function (also not for $\gamma > 0$).
> Our results hold for any loss that satisfies the described condition.
> Hence, we only make a weak assumption on the loss, which makes our results in fact stronger.
> 2. It is not clear whether Theorem 1 also holds for regularized loss.
> It might still hold as long as the regularization does not prevent building levees as we do in our construction.
> 3. For unbounded dimension, the problem is already known to be NP-hard even for a single hidden neuron.
> In fixed dimension, the problem becomes polynomial-time solvable if the number of hidden neurons is also constant (but this is not very interesting).
> Since a network with two layers is already a very simple architecture, it might be tough to find a meaningful restriction that allows polynomial-time training.

---

> > ### Comment · Reviewer_pi8Y · 2023-08-14
> >
> > Thank you for your response. I will keep my rating unchanged.

---

### Official Review · Reviewer_T5Uz · 2023-07-11

**Soundness:** 3 good
**Presentation:** 3 good
**Contribution:** 3 good
**Rating:** 6
**Confidence:** 3

**Summary:**

This paper focuses on two layer neural networks with ReLU or linear threshold activations. The authors show that considering the input dimension (dimension of the data as a constant) equal with 2, there is no polynomial algorithm with respect to the number of data (n) and hidden nodes (k, k<n) that decides whether there exist weights that achieve zero training error. The second result indicates that there is no algorithm with dependence $n^{o(d)}$ on $n$ and arbitrary dependence on $k,d$. Furthermore, if $k=4$ the authors prove $W[1]-$hardness with respect to $d$. Similar results are obtained for the case of threshold activations. Finally, they provide an algorithm, independent of $n$ if the target function is convex.

**Strengths:**

This paper is in general well-written. It closes some gaps regarding hardness results for deciding whether there exist weights on a neural network that achieve a small or zero training loss. The authors show that even when the target is zero error, the problem is NP-hard for $d=2$ and thus for any $d\geq 2$ (since we can simply zero pad and thus ignore the rest of the $d-2$ points (Theorem 1). They also prove that for $k=4$ and zero training error the problem is $W[1]$-hard with respect to the parameter $d$. Their results also hold for the case of linear threshold activations. They finally show that for the case that the target functions is convex, meaning the second layer has weights all equal to one, they show that the problem is fixed-parameter tractable (FPT), when the targeted error is exactly zero. It was already known that for the case of non-zero error, positive error the decision problem is $W[1]$-hard.


**Weaknesses:**

Considering the last result (theorem 10) I am not sure what it adds to our understanding, it will be rare to encounter a convex problem a providing an algorithm to decide whether it is solvable it's not very useful, since we know that a convex function will have a global minimum and that it can be found through classical methods like gradient descent. The authors have already acknowledged that this algorithm will not have any practical relevance, but I also think its contribution as a theoretical result marginal.


**Questions:**

Considering Theorem 1, is there some more tight characterization when $d> 2$ and the inputs are not sparse?

**Limitations:**

The authors have addressed the limitations and this work has no negative societal impact.

---

> ### Author Rebuttal · Authors · 2023-08-03
>
> Thank you very much for your review.
>
> We agree that the algorithm for the convex case is unlikely to be useful in practice.
> From a theoretical side, this result is interesting since it gives a partial positive answer to the open Question 2.
> It is thus a step towards resolving this question and shows that the inherent hardness stems from the case when both positive and negative weights occur in the last layer.
> Besides this, the algorithmic ideas might be inspiration for solving other similar tasks.
>
> Answering your question, for $d>2$, the problem is known to be NP-hard even for a single ReLU with non-sparse inputs (see e.g. Goel et al.).

---

> > ### Comment · Reviewer_T5Uz · 2023-08-13
> > **Response to authors**
> >
> > I thank the reviewers for their response. I would like to keep my score.

---

### Decision · Program_Chairs · 2023-09-21

**Decision:**

Accept (poster)

**Comment:**

This paper studies several computational complexity problems in training two-layer neural networks. The results are solid and address important questions related to the computational hardness of neural networks. All reviewers have recommended acceptance.